

# YAC 1.2.0: An extendable coupling software for Earth system modelling

Moritz Hanke[1], René Redler[2], Teresa Holfeld[2], and Maxim Yastremsky[2]

[1]Deutsches Klimarechenzentrum, Hamburg, Germany
[2]Max-Planck-Institut für Meteorologie, Hamburg, Germany

*Correspondence to:* René Redler (rene.redler@mpimet.mpg.de)

**Abstract.**

A light-weight software framework has been developed as a library to realise the coupling of Earth system model components. The software provides a parallelised 2-dimensional neighbourhood search, interpolation, and communication for the coupling between any two model components. The software offers flexible coupling of physical fields defined on regular and irregular grids on the sphere without a priori assumptions about the particular grid structure or grid element types. All supported grids can be combined with any of the supported interpolations. We describe our approach and provide an overview about some of the algorithms we are using and the implemented functionality. The parallel performance is examined with a set of realistic use cases. The coupling software is now used for the coupling of the model components in the Icosahedral nonhydrostatic (ICON) general circulation model.

## 1 Introduction

With coupling, as we understand it in this paper, we focus on the exchange of physical fields between model components formulated on different numerical grids. Here, we concentrate on model components which can be described as ocean, atmosphere, sea ice and alike, and do not address the coupling of processes within these types of components.

In general, the tasks that have to be covered by any coupling software –named coupler hereafter– are the interpolation between source and target elements and the handling of the data exchange between them. The interpolation is usually based on a matrix-vector multiplication. The required coefficients for this multiplication can be provided by an external program or by the coupler itself. A significant part of the computation of the coefficients is the so-called neighbourhood search that determines a mapping between source and target elements. For a more detailed discussion on these aspects see Redler et al. (2010).

With past generations of Earth system models and the numerical grids involved, the neighbourhood search between any pair of grids, source and target, was not critical in terms of CPU time consumption. The implicitly given connectivity between neighbouring cells, low resolution and thus a small problem size, and the very low degree of parallelism did not pose any challenge to provide efficient algorithms to perform this task. With the advancement of new numerical models formulated on irregular grids, a trend towards higher resolution in the numerical grid and the higher degree of parallelism, we now require flexible and efficient algorithms also in the context of coupling.



A variety of software exists in Earth system modelling for this type of coupling. A very prominent software package taking up this challenge is the Earth System Modelling Framework (ESMF, Hill et al., 2004), mainly written and supported by the National Center of Atmospheric Research (NCAR). The software is written in a mixture of C++ and Fortran90. In particular, ESMF allows for a parallel exchange of data. In addition, it offers a rich functionality far beyond the pure coupling as we

address in this paper. For example, with ESMF individual physics routines can be encapsulated which allows further splitting of individual model components. According to the ESMF-Strategic Plan for 2012-2015 the software package encompasses about 600000 lines of code.

Many coupled climate models use the Ocean Atmosphere Sea-Ice Surface (OASIS) coupler maintained by CERFACS in its version 3. One major advantage, which we think made OASIS a success story, is the lightweight user interface whose

implementation requires only very minor code changes in the user code. OASIS3 supports the most common grid types, among them unstructured grids. For the data transfer, complete fields are collected by individual OASIS3 processes. With a quite recent development OASIS3 uses the Model Coupling Toolkit (MCT). This development step now allows for a direct and parallel data exchange between the participating model component processes. In its current version, OASIS3-MCT still requires to perform the neighbourhood search and calculation of interpolation weights on a single process. Alternatively, external tools

like the Earth System Modelling Framework (ESMF) can be used to provide the required interpolation weights a priori.

Further details about those and other important approaches available so far have been assembled in a collection of articles by Valcke et al. (2012). So why we are not going to use an existing software solution and adapt this to our needs? Our target is to create a software framework which allows us to easily add or replace algorithms in order to test new ideas and concepts not only in theory but also in the real life of numerical modelling. Our primary focus for our software development is on the Icosahedral

nonhydrostatic (ICON) general circulation model (Wan et al., 2013) and to a lesser degree on the model components available with the first version of the Max Planck Institute Earth System Model (MPI-ESM1) (Giorgetta et al., 2013). Last but not least we contribute to the development of the Climate Data Operator (CDO[1]) software.

ICON is designed to run on massively parallel systems. As a consequence we require the coupling software to take full advantage of the hardware and to exchange coupling fields efficiently and in parallel. Furthermore, the handling of large grids

on a single CPU may exceed the available memory; parallel algorithms working on local and thus smaller domains seem to be the only way out. The demand to allow the model components in future to change land-sea masks during the forward integration add the requirement for an efficient and therefore parallel on-line neighbourhood search and the calculation of interpolation weights rather than to perform these steps off-line in a pre-process. Last but not least the software has to be sufficiently modular. This is required to be able for example to add new user-defined interpolation schemes to the existing code. These new

schemes can be tailored to the specific elements in use or physical properties required. In addition, a modular design would allow an easy replacement or the addition of alternatives to already implemented algorithms for the neighbourhood search, interpolation, communication, or other tasks. As CDO supports polygons beyond just triangles and quadrilaterals provided that the grid definition follows the CF conventions, we require algorithms to work with a variety of polygons as well.

---

[1]https://code.zmaw.de/projects/cdo





None of the exisiting software packages offers a combination of parallelised and efficient algorithms in particular for the neighbourhood search and interpolation, an user interface which minimises the interference with the existing model component code, all packed in a concise software package. Several of these features mentioned above have been addressed by OASIS4 (Redler et al., 2010) which is similar in functionality to OASIS3 but with an improved support for the parallelism of climate model components. OASIS4 is able to perform a quite efficient parallel on-line search and to handle the parallel data exchange. However, in its current status, this software does not provide any support for unstructured grids and is restricted to quadrilateral elements. Furthermore, the development and support of OASIS4 has been stopped. Therefore, we started a new approach and created a software framework that fits our specific purpose and allows us to easily use or integrate existing solutions where available. More important, our new framework allows the reuse of successfully tested algorithms and routines within other software projects like the CDO.

We start in Sec. 2 with an evaluation of OASIS4 and discuss pros and cons that drive our design decisions for the new software. Our new approach is introduced in Sec. 3. We dedicate Sec. 4 to describe some key features of the user interface before we investigate the performance of the software in Sec. 5 and further discuss and compare our approach in Sec. 6. We finish this paper with some major conclusions in Sec. 7 and provide an outlook on future work.

## 2  Lessons learned from OASIS4

Two of the authors were deeply involved in the development of the OASIS4 coupling software. The experience gained with OASIS4 is the basis for our new coupling solution.

Within the Programme for Integrated Earth System Modelling (PRISM, Valcke et al., 2007) the development of OASIS4 started from scratch in 2001. It continued with institutional funding until 2009 and some more evaluation and maintenance within the Infrastructure for the European Network for Earth System Modelling (IS-ENES, Joussaume, 2009) project until 2010. The main objective of OASIS4 was to provide an efficient parallel coupling solution with online regridding capabilities to couple climate model components via minimal intrusive interface routines.

OASIS4 consists of two independent software components, a library that is linked to all model processes and a coupler application. The library is responsible for determining which source data is required for the interpolation of each target point or cell. This neighbourhood search is done during the initialisation phase. For most interpolation methods the search is the most time consuming task and requires a lot of communication. Concurrently to the model processes an arbitrary number of coupler processes has to be started. Based on the results of the neighbourhood search, these processes compute the interpolation weights, do the actual interpolation and serve as a data broker between the different models. The coupling setup is defined by the user via XML files which can be generated through a Graphical user interface. The programming language for OASIS4 is mainly Fortran90 with some few parts written in C. For a more detailed description of the parallel search algorithm and other particular aspects of OASIS4 the reader is referred to Redler et al. (2010).

A major strong point of OASIS4 is its neighbourhood search due to the very efficient parallel multi-level search algorithm. In addition, the data exchange between the models which includes the regridding is also very fast. In the setup phase commu-



nication between source and target processes is handled asynchronously, a prerequisite to achieve good load balancing and to allow for overlapping communication and computation.

The primary focus of the OASIS4 design has been put on performance aspects. A key issue in terms of memory and CPU time is to take advantage of the a priori knowledge about the numerical grids. As a result special grid structures like regular
grids in longitude and latitude have been supported first, followed by curvilinear block-structured grids. For each new grid type huge parts of the code have to be adjusted or extended to accommodate for the specific characteristics of the new type. This makes the software stack growing almost exponentially and the software harder to maintain and debug.

The concept of independent coupling processes that are always ready to send, process and receive data showed in practice no benefit and only increases complexity. Another problem of OASIS4 is the lack of unit tests. Testing is solely done using
simple toy models. Thus, extensive testing of parts of the software is not possible.

## 3 YAC - Yet Another Coupler

"A complete rewrite of legacy science software, with a specific focus on clarity, can prove useful. It uncovers bugs, provides a definitive statement of the underlying algorithm, lets scientists easily pose and answer new research questions, makes it easier to develop new visualisations and other interfaces, and makes the code considerably
more accessible to interested third parties and the public." (Barnes and Jones, 2011)

YAC started as a small software project which we used to gain practical experience with a variety of methods, programming paradigms and software tools which are already well known but have not found its way into the climate modelling community on a broader scale. Rather than doing this in the context of an abstract software project we chose the coupling for Earth system models as a real use case. We have taken this opportunity to generate a framework which allows us to test new concepts and
algorithms. Our goal here is to be able to easily replace algorithms or add them for performance comparison purposes and to allow for an easy extension later towards a richer set of functionality.

As already outlined before we favour the OASIS-like approach due to its user-friendliness concerning the specification of the Fortran API and later use in a coupled Earth system model. In contrast to other existing software solutions this allows us to meet other boundary conditions posed by the ICON project. Despite its advantages the OASIS software suffers from several
drawbacks that hamper its use in our case. With YAC our aim is to inherit successful concepts from OASIS4 (but not software in terms of source code) and at the same time find alternatives were we consider OASIS4 to be less flexible. A modular design allows for other software tools like the CDOs to benefit from subsets of the provided functionality. A subset of internal YAC routines has already found its way into recent CDO versions. With the current version of YAC we provide different algorithms to calculate areas on the sphere which are enclosed by polygons. Tests are available to check the quality of the results for
different cell types which allows us to identify the most stable algorithm or replace the selected one by a new alternative. In the same way different search algorithms can be implemented and tested against each other with respect to performance and scalability.



## 3.1 Best practices

Best practices of code development found its way only recently into the literature of climate science (Clune and Rood, 2011; Easterbrook, 2012). Being a small group without formal management we did not set up a formal document about a coding style, but we nevertheless defined a set of rules and principles for our development work in order to streamline and focus our
effort.

### Programming language

One of the most fundamental choices is the selection of an appropriate programming language. We decided to use C for the following reasons: In our particular case we are convinced that most parts of the software are much easier to program in C rather than Fortran. Specific examples are the allocation and reallocation of memory; in general C is more flexible in the handling
of dynamically allocated memory. Non-commercial debugging tools for C are in general more mature compared to those available for Fortran. Debugging of Fortran programs with the GNU debugger (GDB) can be more difficult with some Fortran compilers. For example module subroutines get automatically renamed and the inspection of data structures can sometimes cause problems. In our opinion the same argument holds for Valgrind [2] which helps during the development to detect memory leaks in a quite early stage. Writing portable code is far easier compared to Fortran as there is less diversity among C compiler
than it is the case with Fortran. Especially the different levels of Fortran standards give the programmer a tough job to write portable code. As already indicated by quoting Barnes and Jones (2011) at the beginning of this section using C makes it much more attractive for a larger class of programmers outside the climate modelling community to join this effort and thus helps to bring in different expertise.

The CDO software in many aspects has requirements quite similar to a coupler with the major difference being that the
transfer of data happens between files rather than model components. As in its current implementation the CDOs use in parts the same algorithms for the neighbourhood search and calculation of interpolation weights as OASIS3, they also suffer more or less from the same lack of performance (when data have to be processed on global high resolution grids) and interpolation errors close to the poles. The CDOs are programmed in C, and thus the CDO software can directly benefit by using parts of YAC.

Since Earth system model codes such as ICON (and the majority of other climate model code) are written in Fortran we also provide a respective interface.

### Test suite

We provide short test programs that serve two purposes. The test programs demonstrate how to use particular subsets of the functions and immediately explain the general usage (interface) and the context in which a particular function is intended to
be used. Thus, the test programs itself already serve as a documentation of our software. With the help of these test programs it is easier to debug the code and to detect programming bugs already at early stages during the development process as these

---

[2]http://www.valgrind.org





short test programs can be run at a high frequency. Furthermore the tests allow to systematically check for most special cases. Most of the test programs have the character of a unit test which forces us to keep the different parts of the code independent with well defined interfaces. Overall, the tests cover a large portion of the code and quickly highlight unintentional bugs due to unknown dependencies. In addition to these short test programs we provide a set of examples which focus on the usage of the

Fortran and C user API. These simple toy models demonstrate the use of the API, and the code sections can be transferred into real model code.

**Documentation**

Even though a proper documentation of software is key to any software project it is often neglected because programmers consider it an unattractive and time consuming task. It is very challenging to keep external documentation up to date with

the software development unless you have a large development team with sufficient resources to dedicate some of these to documentation. In our case we rely on Doxygen[3]. Having the source code documentation and the source code itself in the same place eases the task of keeping both parts in synchronisation with each other. Our main repository is automatically checked for new commits in regular intervals. If new commits are found the YAC Doxygen web site[4] is rebuild, which guarantees the availability of an up-to-date documentation at any time. As the Doxygen configuration file is part of the software users are able

to generate a local copy of the html tree.

**Style guide**

As indicated above we have not provided a written style guide. Nevertheless we stick to certain coding rules which we established while coding the first routines. The principle is then not to deviate from the once established style anymore. We use long names for routines, data types and variables which makes the code longer but more readable and easier to understand.

Wherever possible we avoid global variables. We restrict access to contents of internal data structures by using information hiding schemes. Access to these data structures is only available through a set of interface routines. We elaborate further on this in Sec. 3.1. Compared to OASIS4 we have kept the individual functions shorter, which again increases the readability of the code.

**Version control**

We use the version control system git. The main advantage for us is that git allows for smaller (local) commits during the development of a special feature which are independent of the availability of any network access. Only when a particular development cycle is finished the changes are pushed to the remote directory that is accessible to all, which then also generates a single notification by email. The way git is handling the branches and especially the ease of switching between branches we feel very comfortable with.

---

[3]http://www.doxygen.org
[4]https://doc.redmine.dkrz.de/YAC/html/index.html





**Optimisation**

We first concentrate on the development of functionality without a particular focus on optimisation. Highly optimised code often comes at the cost of reduced readability, and the resulting code is often less flexible to changes of the algorithm. For some parts of the workflow, we deviated from this paradigm, because based on experience gained from OASIS3 and OASIS4 we know that high performance is key for certain aspects. Therefore, we implement efficient algorithms for the search (see Sec. 3.3) right from the start. The modular structure of the code facilitates this work as algorithms and even data structures can be replaced without any impact on other functions. We have already minimised the number of messages by packing data. Apart from some special support for completely regular grids in longitude and latitude we do not make use of any implicitly given connectivity as it is the case for block-structured grids. Rather they are treated internally like any other unstructured grid, and we see some potential for optimisation in order to speed up the search on this type of source grids.

**Object-oriented programming**

Even though the implementation is done in C, we have tried to use an object-oriented programming paradigm. Most data is stored in objects, which are C-structures that contain a number of related data fields. The data within an object is normally not accessed directly. Instead, it is read or modified using a set of interface routines.

The major advantage of this approach is that different parts of the code only depend on the interfaces of each other rather than directly on the actual data. By using the interfaces all access to the data is controlled by the code that is directly associated to the data. This avoids misuse of its content and unwanted side effects that might happen when changing seemingly independent parts of the code. In addition, this allows changing of the data structures while keeping the original interface. Such a change might be required in case an algorithm is modified or replaced.

One example of such an object within YAC is the grid. YAC supports multiple grid types. Each grid type has its own method of storing its data, such that memory consumption and access is optimal (e.g. regular grids do not store explicit connectivity information). However, there is one set of interface routines which supports all grid types. All algorithms in YAC access grid data through this interface. As a result, parts of the code that use this interface work with all grid types. For example a new grid type would automatically work with all existing interpolation methods.

The interpolation method object is another example. In YAC an interpolation consists of a number of arbitrary interpolation methods. An interpolation starts by applying the first method to the data that is to be interpolated. Data points that cannot be handled by an interpolation method are passed to the next one. This is only possible because interpolation methods have a common interface, which is used to pass the data between them without knowledge of the other interpolation method type. The interpolation and a practical example of applying a sequence of interpolation methods is presented in Sec. 3.4

## 3.2 Communication

YAC can be considered as a kind of abstraction which hides the complexity of the Message Passing Interface (MPI) for the communication between model processes. Like MPI, YAC is programmed as a pure library which has to be linked to the



model code. YAC enables the model component processes to communicate directly with each other depending on the coupling configuration.

Internally, YAC does all communication through its own communication layer. This allows us to do the communication inde-
pendently of underlying low-level communication routines (e.g. MPI). In addition, this layer is used to enhance the capabilities provided by the MPI library. Currently, there are two implementations of this layer. The standard one is based on MPI. For debugging purposes there is also a second implementation of the communication layer that simulates a parallel environment within a single process. This allows us to simulate a parallel application that uses asynchronous communication without the effects of non-deterministic behaviour.

In the MPI variant we avoid blocking operations in favour of an asynchronous communication scheme. For the sending we inherit the non-blocking buffered send from OASIS4 with its internal buffer management. For the receiving counterpart we provide a modified version of the standard asynchronous receive operation. Instead of the request argument, our implementation requires a callback function and user-defined data pointer. This callback function is called once the respective message is received by the communication layer. The received data and the provided data pointer are passed as arguments to this function.

We use these communication routines to split the complete workload into tasks. Each task has its own callback routine. By calling the asynchronous receive routine and by passing the respective function pointer a task is "activated". Once the respective data has been received by the communication layer, the task is processed. Each task can trigger other tasks by sending the respective messages. This generates dependencies between tasks. All tasks and their dependencies can be seen as a directed acyclic graph (DAG). The main advantage of this communication scheme is that independent tasks can be processed in
any order, which should result in a good load-balancing. OASIS4 has a similar functionality. However, the YAC implementation is by far more general, which allows usage of this communication scheme by independent parts of the code without interfering with each other.

### 3.3  Search

During the initialisation phase a so-called global search is performed. This is done once for each pair of source and target grids
for which any interpolation is required. The grids can be distributed among multiple processes, and the global search does a basic intersection computation between all cells of both of these distributed grids. The result is a prerequisite for all interpolations. It greatly simplifies the interpolation specific search described in Sec. 3.5. Furthermore, it reduces the communication between source and target processes in the interpolation search step.

In order to emphasise that we support a large variety of cells we call these *polygons* from now on. For the global search we
require that within a grid all vertices of the polygons and the polygons itself are unambiguously identifiable by ids. For grids that are distributed among multiple processes the parts of the grid on each process need to have a halo with a width of at least one polygon. Each vertex, edge or polygon must be owned by only one process. In case of halos the user needs to provide the rank of the owner process of the respective data points.

Since computations of intersections between polygons can be computationally intensive, we use bounding circles as a basic estimate for possible intersections. On the sphere bounding circles have better properties than bounding boxes. A particular



advantage of circles is the computation of the intersection between two of them on the sphere, because it is simple to program and fast to calculate. Furthermore, our experience with OASIS4 has shown that bounding boxes in the longitude - latitude space can have problems at the pole regions of the sphere.

If the computation of the intersection of the bounding circles yield a possible intersection between the polygons, the exact result needs to be computed. A basic operation required to do this is the computation of the intersection between two polygon edges. We currently differentiate between three different edge types: (1) edges that are located on great circles, (2) circles of longitude, and (3) circles of latitude. Intersection computation involving great circles is done in 3D-Cartesian space. In the pole regions of the sphere this is much more accurate than using trigonometric functions in the longitude - latitude space.

To reduce the number of checks that need to be done in order to determine all intersecting polygons of both grids, we have introduced an intermediate step. For this intermediate step we implemented two different algorithms: a bucket- and a tree-based search algorithm.

**Bucket-based search:**

Our first approach was a bucket search. It starts by mapping cells of the source and target grid to an intermediate grid that
is relatively coarse compared to the source grid. We use a Gaussian grid as the intermediate grid. Due to the properties of this grid, the mapping operations have a low computational cost. Afterwards, it is sufficient to compute the intersection between source and target cells that are mapped to the same cell of the intermediate grid.

The bucket search suffers from the fact that the "cell densities" of the intermediate grid and the source and/or target grid can deviate significantly. This can result in a high number of source and/or target cells being mapped to a single cell of the
intermediate grid. As a result more source and target cells might have to be compared to each other than necessary.

**Tree-based search:**

As an alternative to the bucket search a search tree was implemented. Based on the bounding circles of all cells of the source grid a search tree is generated. Using this search tree, we can then find all source cells whose bounding circles overlap with bounding circles of the target cells. Since the algorithm is based on the bounding circles of the grid cells, all computation is
rather cheap. In our tests we have seen that the tree-based search algorithm is more than twice as fast as the bucket algorithm. Therefore, this is now the preferred algorithm in YAC.

### 3.4    Interpolation stack

In a typical coupling setup where a data field is sent from a source component to a target component, the grids of both components are not identical. This is particularly true for land-sea masks. Therefore, a target may not be completely covered
by valid source polygons such that a particular interpolation may not be suited to provide data for this target. In order to provide a solution to these problems in YAC all interpolation methods are independent of each other but share a common interface. Multiple interpolation methods as listed in Sec. 3.5 can be combined in the so-called interpolation stack. This allows the user to freely select any sequence of interpolation methods individually for each coupling field.

In Fig. 1 we apply an example of such an interpolation stack for interpolating the WOA09 sea surface salinity onto the ICON R2B4 grid, visible as grey triangles. As the primary interpolation we choose the 1st order conservative remapping (Fig. 1a).





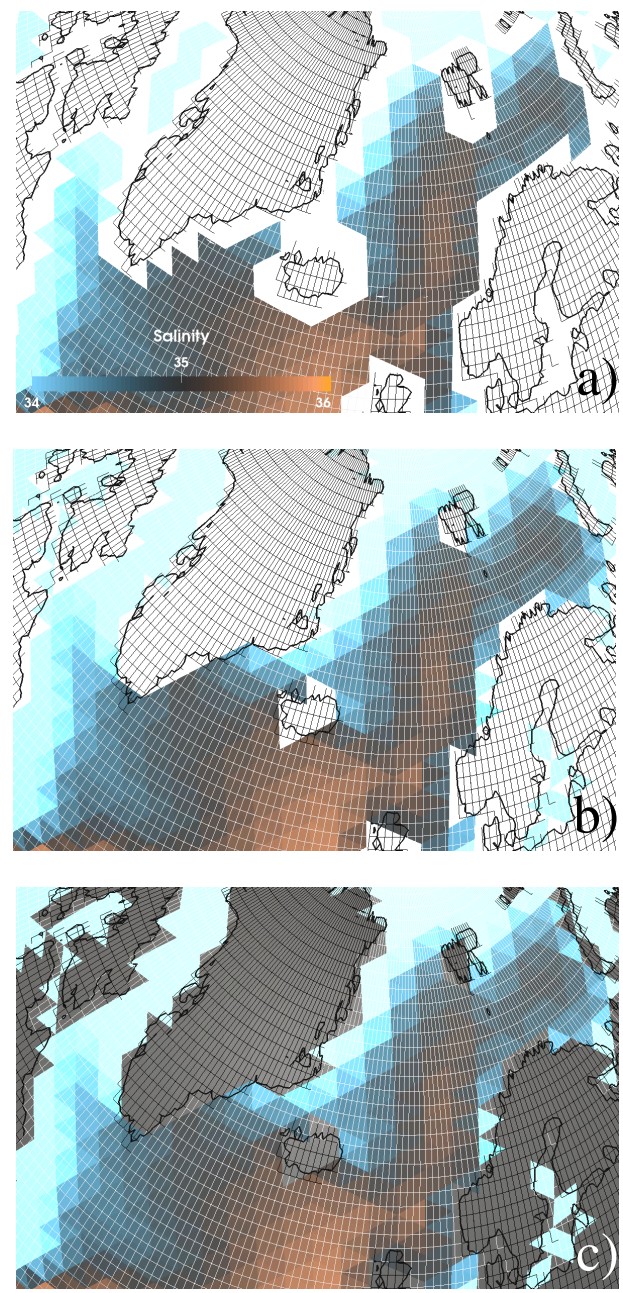

**Figure 1.** Interpolation of WOA09 January monthly mean sea surface salinity (for details see Sec. 3.4)

Here only those target cells get a value which are completely covered by valid (non-masked) target cells. In a second step we try to interpolate remaining cells using the patch recovery method with a 1st-order polynomial (Fig. 1b). In the third step all



remaining cells which still have not got any value are set to a fixed value, 999 in this case and appear as dark grey areas over continents and narrow bays in Fig. 1c. We note that salinities with values of less that 34 are colored in light blue.

The interpolation stack can easily be modified via the XML interface (see below) to become e.g. a 1st-order conservative remapping followed by a 2nd-order polynomial patch-recovery, and then by a 4 nearest-neighbour interpolation.

We provide a small set of parameters to configure the individual interpolations. The complete list of parameters is described in the GUI user manual which is provided as a supplement to this publication.

### 3.5 Interpolation

An interpolation starts by distributing the target points, which need to be interpolated, among the source processes. For this the global search provides on the target processes for each source process the list of local polygons that have an overlap with the polygons of the respective source process. On the source side, for each target polygons that overlaps with the local domain, a list of all source polygons matching the target polygon is available. This information serves as the starting point for almost all interpolation-specific search steps.

**1st-order conservative remapping**: Using the information provided by the global search, most of the work required for the 1st-order conservative remapping is done. The remaining task here is to determine for each target point a responsible source process (e.g. the process with the lowest ranks working contributing to the interpolation result of the respective target point) and to collect information for those target polygons that overlap with more than one source domain (see First Interpolate Then Transfer (FITT) pattern, Ji et al., 2014) and to calculate the partial overlaps. Different algorithms for the calculation of a polygon area on the sphere are implemented (see area.c and test_area.c for a comparison of these algorithms). Like in ESMF as default we are following L'Huilier's theorem.

**Patch recovery**: Inspired by ESMF we provide the so-called patch recovery interpolation (Zienkiewicz and Zhu, 1992). In our implementation this method is built on the very same set of preliminary information that is available for the conservative remapping. For one particular target polygon we consider all overlapping source polygons for the patch, very similar to the 1st-order conservative remapping. In addition we allow the user to extend this patch by one more row of source polygons surrounding the primary patch. For each source element in the patch we determine sampling points in analogy to Gauss points for a unit triangle and then apply a polynomial fit over these sampling points. The user can choose between 1st, 2nd and 3rd-order polynomials, furthermore the density or number of sampling points can be chosen. The system of linear equations which we have to solve for the polynomial fit can be rewritten in such a way that we are able to calculate a single value (weight) per source polygon in the patch which is only dependent on the geometry. We provide the details about the math on our Doxygen web site. In analogy to the 1st-order conservative remapping these weights are constant in time and can be applied to the physical fields later during the exchange.

**Nearest-neighbour**: Points that need to be interpolated can be initially assigned to multiple source processes. For the nearest-neighbour search we first start by assigning each target point to only one source process. Based on the global search data, we then determine for each target point the n closest source points in the local data, with n being prescribed by the user through the XML configuration file. Afterwards bounding circles are constructed that have the target points at their center. The diameter of




these circles is chosen such that all points found for the respective target points are within the circle. In case a halo polygon is within a bounding circle, a search request is issued to the actual owner of the halo cell containing all relevant information. It can occur that a search request is forwarded multiple times until the final result is found. In case the number of found source

points is lower that n, the bounding circle is iteratively enlarged using the connectivity information until enough points have been found. The n geographic inverse distances are calculated one the sphere. The inverse distances are normalised and stored as weights. As an alternative simple weights with values 1/n are stored to provide the arithmetic average.

**Average**: For the average interpolation we take the source polygon which contains the target point and compute the arithmetic average from all vertices of the polygon. We also provide the option to apply inverse distance weighting to the source vertices

to do the interpolation.

**Precalculated weights**: The file interpolation is able to read in NetCDF files that contain weights. Currently, it only supports a file format that is a simplified version of the weight files generated by the SCRIP library (Jones, 1998). However, it has the potential to be able to handle weight files generated by other programs (e.g. ESMF).

The common use case for applying this interpolation method is to compute the weights once and then reuse them in order to

save time in the initialisation of the model run. This approach is only feasible when reading in and distributing the weights is faster than computing them. Measurements show that computation of the weight at the start of the model run is not necessarily a significant performance factor (see Sec. 5). It depends on the number of processes and the number and complexity of the interpolations used.

Due to our concept of the interpolation stack (see Sec. 3.4) another potential use case is conceivable: If there are target

points that require a special handling, which is not covered by any interpolation method currently available in YAC, the user can provide a file that contains the weights for only these points. These weights could be tuned by hand. The remaining points could then be interpolated using any standard interpolation or even another weight file.

**Fixed value**: Last but not least we provide the option to assign a user-defined fixed value to target polygons or -points which is particularly useful when selected as part of the interpolation stack.

## 25   3.6   Weight file generation

YAC is able to write the weights generated by an interpolation stack to a decomposition independent weight file, which is supported by the file interpolation method (see 3.5). To activate this, the user has to specify it in the XML configuration file. Currently, this is supported for any interpolation stack, except for stacks containing the fixed interpolation (see 3.5). Fixed interpolation is not supported because instead of weights it only generates target point lists that have to be assigned a certain

value. And this type of "interpolation" is usually not covered by typical weight file formats. However, it could be added if necessary.





## 3.7 Cells intersection computation

For 1st-order conservative remapping the coupler needs to compute the area of the overlap region between intersecting cells. It basically consists of three subtasks: (1) computation of the intersection point between edges, (2) clipping of two polygons, and (3) computation of the area of the overlap region.

Most couplers support edges that are represented by a section of a great circle on the sphere. If not identical, two great circles intersect twice. Usually, only one of both points is of interest to the coupler. There are three basic methods to compute this intersection point. The simplest method assumes that all edges are straight lines in latitude-longitude space, which makes intersection computation rather simple. For longitude and latitude circle edges (edges of cells of a Gaussian grid) this is accurate. However, for great circle edges this gets more and more inaccurate the closer the edge is to a pole. A second method uses trigonometric functions to compute the intersection point of great circle edges. Theoretically, this method should be very accurate, but due to numerical inaccuracy we observe problems when computing intersections close to the pole. The SCRIP library (Jones, 1998) provides the option to apply a coordinate transformation (e.g. Lambert equivalent azimuthal projection) for cells that are close to the pole. This is controlled with threshold latitudes. All cells that are above (for the northern hemisphere) or below (for the southern hemisphere) this latitude are transformed. The SCRIP User Guide provides an example threshold value of +-1.5 (in radian). Even though the transformation improves accuracy, it also generates a discontinuity. When using a similar approach in OASIS4 we observed problems in some very rare cases, because it generated "holes" in the grid; some parts of the sphere seemingly were not covered by the global grid. In YAC we apply vector operations in three-dimensional space to compute all intersections involving great circle edges. This method is much more robust and does not require any special handling of close-to-pole cases. In addition to great circle edges, YAC also explicitly supports latitude and longitude circle edges. Unfortunately, latitude circle edges introduce the possibility that two intersection points exist, which can be both within the bounds of the edges involved. This can occur when a great circle edge intersects with a latitude circle edge. The computation of these points itself is not an issue, but it makes the clipping more complicated.

The clipping of two polygons computes the intersection between them. Typical clipping algorithms assume that all edges of the polygons are straight lines. Due to the type of edges supported by YAC, this is not the case here. We tried to approximate the latitude circle edges using sections of great circles, but this increases computation time and makes the cell concave even though in latitude-longitude space it is convex. Currently, we use a modified version of the Sutherland-Hodgman clipping algorithm (Sutherland and Hodgeman, 1974). This algorithm requires that one cell is either a convex cell that has only great circle edges or a rectangular cell consisting of latitude and longitude circle edges. The second cell can be either convex or concave and can have any combination of edge types.

To compute the area of a spherical triangle consisting of great circle edges there are two basic formulas: Girard's Theorem and L'Huilier's Theorem. Originally, we used Girard's Theorem. While testing we noticed an increasing error with decreasing cell size. L'Huilier's Theorem is more complex but yields better results.

The output of the clipping is a polygon that is potentially concave and may contain all edge types. In a first step we assume that all edges of the polygon are great circle sections. We split the polygon into triangles and compute their area using the





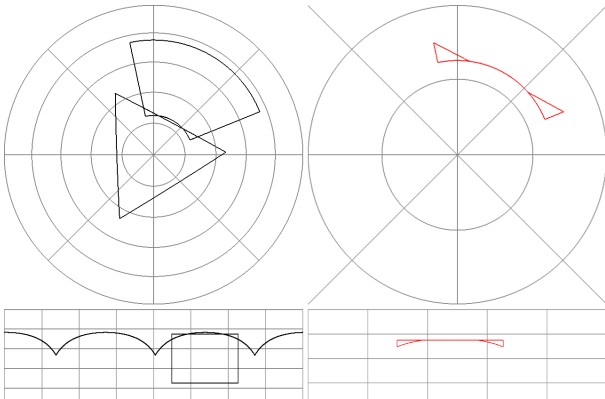

**Figure 2.** Clipping of ICON triangle and ECHAM rectangle

above mentioned theorem. In a second step all latitude circle edges are handled. For each of these edges the error that is made when assuming them to be a great circle edge is computed and either added or subtracted from the overall area depending on the respective case.

Figure 2 depicts a special clipping case near a pole. The triangle consists of great circle edges and is directly on the pole. The rectangle is a typical Gaussian grid cell consisting of latitude and longitude circle edges. The upper part of the figure shows a three-dimensional view directly from above the pole. The lower part depicts the same case in the two-dimensional "Lon-Lat-space". The result of the clipping is a concave polygon with all supported edge types.

To illustrate the difference between great circle and latitude circle edges in the area computation, we have compared the area

of rectangular cells that only differ in the type of the edges. The reference cells are comprised of longitude and latitude circles edges (typical Gaussian grid cells). The other cells have great circle edges. For high resolution grids (edge length of around $0.5°$) the area difference is up to 1 ‰ for cells close to the pole. The area difference is higher for low resolution grid (edge length of around $5°$) where the error goes up to 71 ‰.

## 4   User Interface

Like it is the case with OASIS, our user interface is split into two parts. First we provide a set of library function calls that are to be called from the user code. Using these interfaces, the user makes static information, which is "known" by the model component at run time, available to YAC. This is required to perform the search and data exchange. To allow for some flexibility to explore the functionality of the coupler, a set of free parameters has to be provided at run time like the activation and deactivation of coupling fields or a particular choice of an interpolation or the definition of an interpolation

stack. In OASIS3 this information is provided via the namcouple file, an ASCII formatted file, while in OASIS4 this kind of information is provided via XML formatted files.



## XML

In the recent literature (Ford et al., 2012, and articles therein) it has been pointed out that XML is the preferred language to describe metadata. Similar to OASIS4 we use XML files for the description of the coupling. In the coupling XML file the

user can specify the fields that have to be coupled. Other metadata are provided in the same file as well, like the interpolation sequence and the coupling frequency. In order to facilitate the generation of the coupling XML file a Java based Graphical user interface is available.

A drawback of the design of the configuration XML of OASIS4 is the high complexity of nested elements. Each element representing one couple contains a lot of sub-elements which includes information that in our opinion exceeds the XML

purpose of describing the mere coupling, but instead describes some general properties of components and transients (physical coupling fields). This makes the XML instances hard to read and maintain. The aim of redesigning the coupling configuration XML structure is to simplify and reduce it to the elements necessary for coupling. The resulting new XML structure is much simpler, more readable, and smaller in size.

For defining the structure of XML elements, we provide an XML Schema Definition (XSD). It describes the elements,

attributes, types, keys, and key references (Fallside and Walmsley, 2004). The XSD allows to validate the coupling XML instances. Furthermore, it helps to adjust programs that access the XML files, since it ensures that the XML elements satisfy a predefined structure.

We provide a component XSD and a coupling XSD. An XML instance of the component XSD defines the element structure for basic information about components: model name, transients, grids, and timing information. It is used as the input for our

Graphical User Interface (GUI). The GUI simplifies the generation of an XML instance of the coupling XSD, which provides detailed information on the actual coupling like which transients are to be coupled, the configuration of the interpolations, and the role of the transients, whether they act as source or target.

## Graphical User Interface – the XML GUI

With the design of a minimised XML Schema, the complexity of XML instance files is reduced in a way that it gets well

readable for humans. Still, it is a tedious task to generate XML files by hand and to get all references correct. To ease the editing process of the coupling configuration, we provide a Java GUI that allows the user to create and manipulate the coupling XML configurations with a few clicks. The user can load component XML instances, couple them together, manipulate all coupling configuration settings, and produce a valid coupling XML file. For the implementation of the GUI we chose Java Swing because it provides a simple means for the creation of lightweight platform-independent, portable GUIs (Eckstein et al.,

1998), and because Java provides libraries for easy XML parsing and file I/O.

The initial GUI is showing a pane that is split into 2 parts, left and right. The user can load a component XML to each side, and then couple the transients together. An arrow indicates the coupling direction. With a click on a couple, the GUI presents a detail window where the user can configure the interpolation settings, the time intervals, time lags, and some debug settings.



Figure 3 shows a screenshot of the GUI in action. We provide a more detailed description of the GUI in our manual which is available as a supplement to this publication.

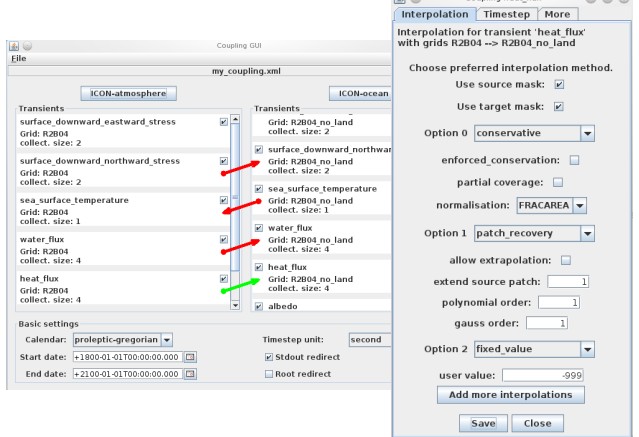

**Figure 3.** Coupling GUI with detail window.

## Application Programming Interface - API

With our API we closely follow the philosophy of OASIS4. For a more detailed discussion about the main principle the reader is referred to Redler et al. (2010), for a detailed description of the YAC API the reader is referred to our Doxygen web site for YAC. As it is the case for the internal library routines the API is programmed in C. As ICON and most other climate codes are programmed in Fortran we provide a Fortran wrapper for the API. In order to get rid of the problem with underscores in symbol names which are handled different by individual Fortran compilers we rely on the ISO_C_BINDING module which was introduced with the advent of the Fortran 2003 standard. The Fortran interface uses overloading with respect to data types and certain data structures. The Fortran interface accepts geographical grid information (longitudes and latitudes coordinates) as well as coupling fields in `REAL` or `DOUBLE PRECISION`. As the C API only accepts `double` Fortran `REAL` is internally converted to `DOUBLE PRECISION` before being passed to the C API. In Fortran grid coordinates for regular grids in longitude and latitude are passed through the same interface routines (due to overloading) as the grid data for unstructured grids while in C different interfaces are provided for the different grid types. We do not provide any direct access to our internal data structure. Instead, we hand back opaque handles (integer values) for components, grids, masks, and fields.

## 5 Performance

We developed two toy models to do performance measurements: perf_toy_icon and perf_toy_cube. The toy model perf_toy_icon is based on an unstructured grid that consists of triangular cells. The other toy is based on a cubed sphere grid. Both toy models cover the whole sphere. Each toy model has one component, which defines an output field that is coupled to an input field





defined on the component of the respective other model. Thus, each component acts as source and target component, and consequently the search is performed within each (source) component.

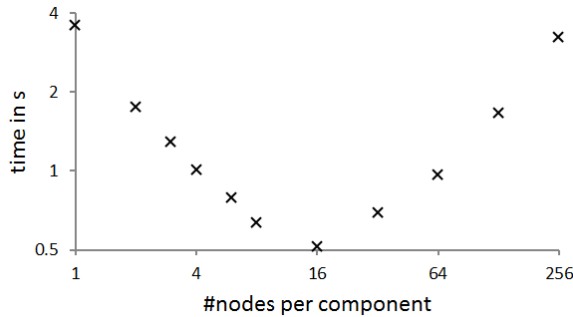

**Figure 4.** Performance of YAC for 1st-order conservative remapping

Figure 4 shows scaling measurements done on mistral at DKRZ, which at the time of testing was equipped with two Intel Xeon E5-2680 v3 12-core processors per node. We used the Intel compiler version 16.0 with Intel MPI version 5.1.1.109.

The measurements ran using 24 MPI processes per node. The toy model perf_toy_icon used an ICONR2B06 grid, which has 245760 cells and around 35km distance between the centers of neighbouring cells. In a typical atmosphere-ocean coupling setup, the ocean model component usally has a higher grid resolution than the atmosphere model component. To mimic this we used a 405x405-cubed-sphere grid (984150 cells; around 25km distance between cell centers) for perf_toy_cube, which represents the ocean in our setup. The coupling was done using a first order conservative interpolation.

The values in Fig. 4 represent the wall clock time required for the call to yac_csearch. This routine does the complete search, which is comprised of the global (see Sec. 3.3) and the interpolation-specific search and weight calculation. Due to the different resolutions, the total workload generated by this routine differs between the two components. By adjusting the number of processes for each coomponent individually it would be possible to minimise the load imbalance between them. To simplify the experiment, we still used the same number of processes for each one. Our measursments did not show a significant

variance. Therefore, each value in Fig. 4 only consists of five individual measurements.

The scaling measurements can be grouped into two parts, the first group being composed of runs with 1 to 16 nodes per component and the second group consisting of runs with more than 16 nodes. In the first group, the computation time required by the global search and the interpolation-specific weight calculation are the dominant factor for the runtime. This work scales well with the number of nodes. At 16 nodes per component (384 processes per component) it has an efficiency of around 44%

for our test setup. At that point, the computation time is negligible and the MPI communication contributes the preponderant part to the runtime. In this case, it seems to increase linearly with the number of nodes.

Since, the wall clock time for yac_csearch in our test case is still below four seconds for 256 nodes per component, which is a total number of 12288 processes for the whole setup, the bad scaling for high numbers of nodes does not seem to be a problem.



An extrapolation of the measurements leads to around 13 seconds for a potential setup with 1024 nodes per component, which should be acceptable.

## 6   Discussion and Outlook

In the context of describing YAC we provide some insight into the search algorithm we are using, the supported interpolations, the parallelism, and elaborate on some design aspects and best practices. With a first development step we have focused on solving the existing problems and on providing the basic functionality. At the same time it is important to have future problems in mind and design the software such that a later integration of new programming modules is not hampered or blocked. We already went through this exercise when we added the first-order conservative remapping or the patch-recovery interpolation (Sec. 3.5). In a second step we have put our focus on the optimisation, for example by providing an alternative to the bucket search (Sec. 3.3) or by introducing the asynchronous collective communication routines.

YAC comprises of roughly 38,000 lines of source code for the library plus another 32,000 lines of code for unit tests. On our machine, the YAC source code compiles in less than a minute and the unit tests take a few seconds to run. These tests have proven to be very useful during the development and also in the exploration phase. It remains a challenge to achieve full test coverage and to test all possible use cases and all internal functionality. Thinking about reasonable tests and the implementation of those test takes some time. At first this tends to slow down the generation of source code for the library part. But by strictly following the rule of writing tests along with the development of library code we are able to detect programming errors at a very early stage; new development is much less likely to break existing code. We are thus convinced that the unit tests help to significantly speed up the development process.

Overall, we consider YAC as being rather efficient when speaking about the number of lines of code, the compile time and the development cycle. This efficiency allows us to test methods and to experiment within a concise software framework. We are able to try out and compare algorithms and methods. Since most parts are programmed in C we directly contribute with our methods to the CDO development. As the CDO software is used worldwide in climate research, the parts which are already transferred into the CDO. These routines are thus used in a much wider context compared to what we can do ourself with YAC alone.

In Sec. 5 we provide some performance measurements for a typical model or grid configuration. YAC scales reasonably well with the number of processes provided that the local problem size remains sufficiently large. With an increasing number of processes and a thus reduced local problem size the communication starts to dominate the time required to perform the neighbourhood search in the initialisation phase but remains within a few seconds. As an alternative the user can precompute the interpolation weights in a preprocessing step and keep them in files such that the weights are read and distributed by YAC at run-time. However, the parallel reading of the weight files and a proper distribution of the weights is not trivial and involves communication among the processes as well. Nevertheless, we are satisfied with the achieved performance.

We shall note here that the time required to perform the search depends on the actual grid configuration, its masks, and the selected interpolation stack. When the distance between a target point and the nearest non-masked source points is large, the



nearest-neighbourhood search can become very time-consuming. In this case the required search radius might be very large and can contain the regions of multiple processes. This increases communication and computation time. Likewise, the calculation for the patch recovery is very costly when the 3rd order polynomial fit is selected together with the maximum number of fix points per source triangle. Here, reducing the number of fix points, selecting a lower order for the fit, or reducing the local problem size by employing more processes can significantly reduce the CPU time. We cannot provide a simple recipe to select

the best interpolation configuration. Like it is the case with the tuning of model physics for new model configurations the interpolation results need to be checked, and interpolation parameters need to be adjusted when necessary.

The communication scheme described in 3.2 works much better than in OASIS4. Due to the general interface, different parts of the code can successfully use the same mechanism without any knowledge of the other parts. For example, implementing the communication for new interpolation methods can be done without any impact on existing communication related code.

As a disadvantage this communication scheme makes the code in some parts hard to read. For some sections of the code we have improved the understandability by including communication diagrams in the Doxygen documentation.

Right from the beginning we have implemented algorithms from which we expect a reasonable weak and strong scaling behaviour. More sophisticated interpolation schemes like a second-order conservative remapping or a bicubic-spline interpolation can help to improve the quality of interpolation results. A build-in support for the coupling of vector fields greatly simplifies

handling of these fields. Further down the road is the support for changing land-sea masks at run time. In the future, the abstraction of the communication layer might enable a thread-level parallel functional decomposition of the work performed by YAC in the initialisation phase.

## 7   Conclusions

In support of constructing coupled Earth system models, we redesign and develop a coupler from scratch. Our focus lies on the

neighbourhood search, the calculation of interpolation weights and the data exchange between model components. The efficient and fully parallelised algorithms directly support unstructured and block-structured numerical grids. This efficiency allows for the online calculation of interpolation weights during the initialisation phase of each model run. Furthermore, we offer the alternative to read in interpolation weights which are generated offline by the climate data operator (CDO) software. This significantly widens the scope of numerical model configurations for which our coupling software can be applied. The software

is available to the climate modelling community and is used in the next generation Max Plank Institute for Meteorology Earth System Model. As an added value, the intentional choice of the programming language — in this case C — allows us to directly transfer parts of our software into the CDO and thus contribute to its improvement.

## 8   Code Availability

Information about access to our software is provided on our YAC Doxygen web site https://doc.redmine.dkrz.de/YAC/html/

index.html under section code availability.



*Acknowledgements.* We are grateful to our colleagues Jörg Behrens, Hendrik Bockelmann, and Thomas Jahns at DKRZ for very fruitful discussions during the development and adding the configure mechanism (TJ). Uwe Schulzweida took the burden and transferred several of our internal routines into the CDO package and was thus able to do far more testing of our algorithms than anticipated and thus helped us to make our software more robust for production use. We appreciate the luxury of having the freedom by our institutes, DKRZ and MPI-M to do some development outside of any project requirements and without knowing beforehand where this will lead us.



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
