# Peer review of "YAC 1.2.0: New aspects for coupling software in Earth system modelling"

_Geoscientific Model Development, 2015_

## Referee Comment (RC1) · Anonymous Referee #1 · 16 Feb 2016

Paper: YAC 1.2.0: An extendable coupling software for Earth system modelling

Authors: Moritz Hanke, Rene Redler, Teresa Holfeld, and Maxim Yastremsky

**1   Summary**

This paper introduces a new coupler library, named Yet Another Coupler (YAC). It provides a flexible API which can be used to couple multiple components of a climate model together, for example an atmosphere and an ocean component. This library provides interpolation and communication routines which can be used to map fields from one component to another in a way that meets some criteria, for example conservative remapping.
[Figure]

This paper documents the interpolations methods provided, describes the justification for certain software choices (for example using C instead of Fortran), and provides some basic performance data.

Overall, the manuscript is written well, with a handful of minor typos. It describes the capabilities and justifications for YAC well.

I would recommend this manuscript for publication in Geoscientific Model Development after some modifications.

**2 General Comments**

Overall, this manuscript is well written and very readable. It spends a lot of time discussing the capabilities of OASIS3 and OASIS4 outside of the introduction, and using them to justify decisions made for YAC. A lot of this seems unnecessary, and could easily be removed without negatively impacting the manuscript. Including this makes the manuscript seem almost like it is intended to describe those couplers as much as YAC.

The performance results are overly simplistic. With only a single result provided for readers to gauge the performance of the coupler. Additionally, performance results are only provided for the global search portion. While this may be the most expensive part of this coupler, it is also only performed once in a given simulation. It would be useful if the authors provided performance results for coupling steps including the interpolation from one grid to another and communicating the results. Additionally, it would be useful to show a semi-realistic example where the two models that are coupled together are not colocated on the same nodes / processors. It would also be useful to provide performance for the global search for additional interpolation methods, similar to the one-off discussion found in Section 6.
[Figure]

The software package makes it seem like the capabilities provided are new enough in that it is flexible enough to easily modify independent portions of the coupling package, to add new interpolation methods or communication routines.

One major issue with the manuscript is that it is not clearly documented where to gain access to the source code or find instructions for building / running the tests that are discussed in the manuscript. Navigating the doxygen site (as recommended in the manuscript), I have been unable to find instructions for downloading / building YAC, so that part was not tested / reviewed, and I cannot comment on the readability of the source code.

Corrections

Page 2, Line 2: "Earth System Modelling Framework" should be "Earth System Modeling Framework", as it is the name of software package. This change should be made throughout.

Page 2, Line 14: "to perform the neighbourhood search" should be "the neighbourhood search to be performed"

Page 2, Line 17: "So why we are not" should be "So why are we not"

Page 2 Line 17: "and adapt this to our needs?" should be " and adapt it to our needs?"

Page 2, line 26: "allow the model components in future" should be "allow the model components in the future"

Page 2, line 28: "off-line in a pre-process." should be "off-line in a pre-process step."

Page 3, line 27: "coupler processes has to be started." should be "coupler processes have to be started."

Page 3, line 28: "do the actual interpolation and" should be "do the actual interpolation, and"

Page 3, line 30: "mainly Fortran90 with some few parts" should be "mainly Fortran90 with a few parts"

Page 4, line 6: "extended to accommodate for the specific" should be "extended to accommodate the specific"

Page 4, Line 7: "software stack growing almost exponentially" should be "software stack grow almost exponentially"

Page 4, line 8: "send, process and receive" should be "send, process, and receive"

Page 4, Line 17: "paradigms and" should be "paradigms, and"

Page 4, line 17: "but have not found its way" should be "but have not found their way"

Page 4, line 22: "As already outlined before we" should be "As already outlined before, we"

Page 4, line 24: "Despite its advantages the OASIS" should be "Despite its advantages, the OASIS"

Page 4, line 25: "With YAC our aim" should be "With YAC, out aim"

Page 4, line 30: "or replace the selected one" should be "or replace the current one" or "or replace a selected one"

Page 4, line 31: "the same way different" should be "the same way, different"

Page 5, line 14: "among C compiler" should be "among C compilers"

Page 5, line 30" itself already serve as" should be "itself already serves as"

Page 6, line 1: "Furthermore the tests" should be "Furthermore, the tests"

Page 6, line 4: "short test programs we" should be "short test programs, we"

Page 6, line 13: "web site is rebuild," should be "web site is rebuilt,"

Page 7, line 11: "this type of source grids." should be "this type of source grid."

Page 7, line 24: "For example a new" should be "For example, a new"

Page 9, line 30: "may not be suited to" should be "may not be well suited to"

Page 10, figure 1: Please expand the caption of this figure. Alone the caption is not enough to determine what each of the subfigures represents, or what the triangles / rectangles are.

Page 11, line 1: "not got any" should be "no"

Page 11, line 2: "of less that 34" should be "of less than 34"

Page 11, line 11: "for each target polygons" should be "for each target polygon"

Page 12, line 5: "is lower that n" should be "is lower than n"

Page 12, line 6: "are calculated one the sphere" should be "are calculated on the sphere"

Page 12, line 7: "As an alternative simple" should be "As an alternative, simple"

Page 15, line 24: "in a way that it gets well" should be "in a way that it becomes easily"

Page 17, Figure 4 is captioned with "Performance of YAC for 1st-order conservative remapping", while the text says that Figure 4 represents only the global search portion of the interpolation.

Page 17, line 13: "for each coomponent" should be "for each component"

Page 18, line 22: The sentence that begins with "As the CDO software is used world-wide" doesn't make sense. Please rewrite it.

Page 19, line 14: "A build-in support" should be "The built-in support"

---

## Referee Comment (RC2) · Anonymous Referee #2 · 19 Feb 2016

General Overview:

This paper provides an overview of the YAC1.2.0 coupler focusing mainly on the interpolation weights generation. It is reasonably well organized and written and with revisions, will contribute to the literature on ESM coupling.

I am a bit torn about the organization of this paper. The paper focuses on the YAC weights generation, but the title suggests the paper is a description of the YAC coupler. I think this paper should be revised to clarify the focus. I recommend this paper focus on interpolation in YAC, by providing more details about the weight generation algorithms, show more scaling data for other weights generation options, describe the weights generation API in YAC, describe the remapping algorithm in the code and add some scaling data about the remapping cost in YAC. I believe this paper could meet those

goals relatively easily, although some of the current text may need to be removed and refactored. I would also recommend a change in the title in this case.

An alternative would be to turn this into a YAC overview, but in my opinion, that would require significant rework. The paper would need to add more information about hooking up models to YAC including some further overview of the strategy, it would have to discuss how data is passed between models and the strategy employed in communication, lags, coupling frequency management, sequencing, concurrency, and other aspects of the system. It would also need to add further historical description of other ESM couplers in the introduction in order to change the focus.

In many ways, I think both papers above are worth pursuing. This software effort needs to be published. There are several interesting and unique aspects of the weights generation that can stand alone in a paper. A YAC overview paper is also appropriate to describe overall design to the community, especially if YAC is available for use within the community. Unfortunately, I think the current paper is a bit of both and is not properly focused or complete in either case.

Specific Comments:

Overall, I believe there is a shortfall in the number of cited references. If this is a focus on weights generation, there are a number of references that should probably be sited in the introduction and text. If this is a focus on new ESM coupling software, again there needs to be additional references and some additional historical perspective added. Even without changes, references need to be added (for instance, Oasis3 on page 2, line 8 and MCT on page 2, line 12 but there are others). The authors need to make an effort to properly reference work sited in the paper.

Please clarify the following. Are the weights generated on the source or target side, on the union of all tasks or is it up to the user? Is remapping done on the source or target side or as part of communicating data between models? Is the communication of data between models separate from remapping or part of remapping? There are hints of

the answers in the text, but it isn't clear.

The authors are extremely familiar with Oasis4, but it's unlikely the readers will be. It is fine to compare to Oasis4 and point readers to an Oasis4 reference, but it is also important to make sure the description does not assume the readers are inherently familiar with Oasis4 and that it does not require a study of Oasis4 as a prerequisite.

How does YAC ensure that the weights generated on the fly are of high quality? The text notes problems at poles in various algorithms, issues with different types of grid-cells and different edge options, the requirement that one grid have either convex edges or be rectangular. Are those things checked on the fly? Are the weights somehow checked for conservative or gradient properties independently after they are generated? Are the properties at the pole checked? Is this a concern?

Page 1, Line 20: "not critical" is not a proper description. ESMs have been carrying out computations on moderate to high resolution grids for several years and the (pre) computation of interpolation weights in terms of both performance and quality has been a critical issue. This issue has taken up not insignificant resources in several projects including OASIS, ESMF, and SCRIP.

Section 3.2 describes the communication implementation and is far more important than much of the other material before it. I would like the description here to be expanded a bit, especially as related to lines 10-22. You need to describe "the non-blocking buffered send from Oasis4" for those that are not familiar. Please explain how the callback and data pointer work in a bit more detail. This is not entirely clear. Remember that your audience didn't work on either the Oasis4 or YAC implementation. Also, maybe there needs to be some further clarification on what aspect is being described. There is communication associated with weights generation, interpolation, and coupling data. Is this communication approach used for all of these?

Section 3.3. The requirement on the user to decompose the grid in a way that also includes a halo region and the rank owner of each halo gridpoint seems to be rather

inflexible. That datatype and information will almost certainly have to be computed specifically to support the YAC weights generation interface and is unlikely to be a natural part of any model decomposition. The halos are probably only needed in a subset of interpolation methods (like bilinear) and maybe YAC should be computing that connectivity, not the model. In addition, in a decomposition like round robin where each process has a random(ish) set of points, the halo description is going to require that "n" halo points be specified for each gridcell, increasing the memory and complexity. YAC would be much more usable if the connectivity were computed within the coupling layer when needed.

Section 3.4. The interpolation stack feature is well thought out and something that other weights generation methods are not able to do easily yet but is needed. Well done.

Section 3.5. Do you have a bilinear interpolation option? This option is heavily used in ESMs.

Section 3.7. Does the current calculation of intersections guarantee conservation. The overlapping areas have to be computed in a way that the partial areas add up to the areas of each grid cell in total. I think page 14, line 3 suggests the areas are handled properly, but maybe a sentence stating this would clarify.

Section 4. I find the description of the user interface a bit out of place. The article is really focused on the interpolation weights generation. Nothing has been presented about how interpolation is carried out nor how models are coupled. This section, describing how to setup YAC via XML does not seem to fit into the paper.

Section 5. The total time and scaling for weights generation is very good. I think it's also important to show similar timing information for patch (1st, 2nd, and 3rd) and nearest neighbor calculations as well as bilinear if it's available. These are very different algorithms and I think providing scaling information for all three is important for this paper. In our experience, the 1st order conservative weights generation is sometimes

[Figure]

the easiest and fastest to carry out. (This is also noted in Section 6, so data to back it up would be very useful)

Figure 4 title seems incorrect. According to the text, it is the time to generate 1st order conservative mapping weights, not to carry out remapping.

Section 6. Line 25. The only thing that has been shown in the text is the cost of the weights generation for 1st order conservative. It is a stretch to say "YAC scales reasaonbly well". Much of the YAC performance is not shown including remapping performance and cost to communicate data back and forth between models, the ability to support concurrency and other issues. Section 6. Line 30. It would be nice to document the actual time needed to write and read mapping files as this is already available in YAC to have a quantitative comparison against the cost of generating the mapping weights. It would not surprise me if the read operation was more expensive than online weights generation for the test case used, and the actual numbers would add to the paper.

Section 8. The web info is pretty useful. Just FYI, I think it is missing a "getting started" type of documentation for new users to know what calls are needed and how to organize them in their models.

Technical Corrections:

The grammar is pretty good in this paper, but could be improved. I would recommend a review by a native English speaker if possible. A few problems are noted below, but the list below is not comprehensive.

Please make sure to have references for both L'Huilier's Theorem and Girard's Theorem in the paper.

Please add a reference to the YAC Doxygen web site the first time it is mentioned in the paper. I know it's list in section 8, but a reference would also be good.

Page 1, Line 20-22, clean up sentence

Page 2, Line 3: ESMF is not NCAR. It's now NOAA. Get rid of the phrase starting with "mainly written..."

Page 2, Line 13: fix "still requires to perform", not correct grammar

Page 3, line 30: change to "with a few parts written in C"

Page 4, line 7: change "growing" to "grow"

Page 5, line 15: remove "it" from "than it is the case with Fortran"

Page 5, line 20-23: sentence is difficult to read

Page 6, line 8: remove "a" from "Even though a proper documentation"

Page 6, line 13: rebuild -> rebuilt

Page 6, line 14: need a comma after "software"

Page 6, line 18: anymore -> thereafter

Page 6, line 28-29: "We feel very comfortable with" should start the sentence, although that sentence is a bit awkward in either case.

Page 7, line 10: remove "it"

Page 11, line 10: need some commas.

Page 11, Section 3.5, paragraph 1, please rewrite to make a little easier to understand.

Page 11, Section 3.5, paragraph 2. I think the main point is to determine which process will carry out the interpolation and how to collect information for polygons that overlap multiple processes on the other grid. Please rewrite to make it a little easier to understand. Also, could you just add 1 sentence about what FITT does?

Page 12, line 6: one -> on

Page 15, line 9: what is "one couple"?

Page 18, line 1: I would remove this sentence. It's dangerous to extrapolate in these cases and the data speaks for itself.

Page 19, line 19: change to "...,we designed and developed a coupler..."
* * *

---

## Author Comment (AC1) · 3 Mar 2016

We would like to thank the reviewer 1 for the careful reading and the detailed review. Please find below our comments on the review.

> *Overall, this manuscript is well written and very readable. It spends a lot of time discussing the capabilities of OASIS3 and OASIS4 outside of the introduction, and using them to justify decisions made for YAC. A lot of this seems unnecessary, and could easily be removed without negatively impacting the manuscript. Including this makes the manuscript seem almost like it is intended to describe those couplers as much as YAC.*

In the "coupling" community, the question arose: why we had to invest into a new coupling software. Therefore, we felt the need to clarify the differences. In addition, OASIS3/4 is our main source of experience regarding coupling. Many of our decisions are based on these couplers, so use them to justify our design and give credit to them. As referee 2 had similar concerns we will shorten the manuscript and concentrate on YAC.

> *The performance results are overly simplistic. With only a single result provided for readers to gauge the performance of the coupler. While this may be the most expensive part of this coupler, it is also only performed once in a given simulation.*

As stated in the paper, we did not really care about the performance of the coupler as long as it was within reasonable bounds. For the paper we put our focus on the design of the coupler itself. With the given performance results we wanted to show that it is working with acceptable performance. To enhance performance results we will do additional measurements with other grid resolutions and other interpolation methods. We will also explicitly measure the time required by the global search and the interpolation computation.

> *It would be useful if the authors provided performance results for coupling steps including the interpolation from one grid to another and communicating the results.*

Our measurements showed that the coupling step itself (including interpolation and exchange) is really fast and hence boring. Furthermore, we do not consider the data exchange as time critical for real coupled applications. Wait time due to load imbalance is typically much more time consuming than the exchange of data. But we may add a respective diagram.

> *Additionally, it would be useful to show a semi-realistic example where the two models that are coupled together are not colocated on the same nodes / processors.*

It seems like the paper was not clear on this part. For example, in the last measurement we used 256 nodes (6.144 process/individual MPI ranks per component (see x-axis label of fig. 4) which adds up to  a total of 512 nodes or 12.288 processes. We will further clarify this in the text.

*It would also be useful to provide performance for the global search for additional interpolation methods, similar to the one-off discussion found in Section 6.*

The global search is done only "once for each pair of source and target grids for which any interpolation is required." (see section 3.3) The global search results are available to all interpolation methods, which in turn do additional communication to fulfil their special requirements.

*One major issue with the manuscript is that it is not clearly documented where to gain access to the source code or find instructions for building / running the tests that are discussed in the manuscript.*

As mentioned in section 8, on the main page of our Doxygen page (https://doc.redmine.dkrz.de/YAC/html/index.html) we have a paragraph about code availability:

**Code Availability**

Tagged versions of the software are available upon request. Please contact Moritz Hanke ( hanke at dkrz.de ) or Rene Redler ( rene.redler at mpimpet.mpg.de ) and provide your name, institution and a few lines describing your intention with yac.

In order to guarantee anonymity for the reviewers we provided a link to an archive file to the Topical Editor. This tar file contains the status of our source code at the time when the paper was accepted for publication in GMDD.

We are very sorry that this was not communicated clearly enough. We will add those details about code availability in the revised version of the manuscript.

*Page 16, Figure 4 is captioned with…*

"The values in Fig. 4 represent the wall clock time required for the call to yac_csearch." (see section 5) This includes the global search and all communication and computation necessary to compute and distributed the weights required for the interpolation.

*Corrections*

We would like to thank the reviewer once again for the careful reading of our manuscript and the suggestions for corrections. We will of course include all of these in our revised version of the manuscript.

---

## Author Comment (AC2) · 3 Mar 2016

We would like to thank the reviewer 2 for the careful reading and the constructive review.

**General Overview**

With this paper we wanted to present YAC with a special focus on a set of selected key aspects that we did differently compared to other coupling solution. The weight generation algorithms we use are more or less the same as in any other coupler and are hence, from our point of view, sufficiently documented. We will try to clarify the focus of the paper.

Overall, I believe there is a shortfall in the number of cited references.

Additional references will be added accordingly.

Are the weights generated on the source or target side, on the union of all tasks or is it up to the user?

The search and the calculation of weights is performed on all source processes (owners of the source grid). Here the data are interpolated onto the target before sending. We will update the revised version of the manuscript accordingly.

Is remapping done on the source or target side or as part of communicating data between models? Is the communication of data between models separate from remapping or part of remapping?

We did not consider our implementation of the remapping step particularly interesting and therefore a detailed description has been omitted.

First, the source processes communicate with each other for a kind of halo exchange in order to provide the necessary data for the processes that have to calculate the stencil. We follow the referees request and will modify the revised version of the manuscript accordingly by adding an additional chapter.

The authors are extremely familiar with Oasis4, but it's unlikely the readers will be. It is fine to compare to Oasis4 and point readers to an Oasis4 reference, but it is also important to make sure the description does not assume the readers are inherently familiar with Oasis4 and that it does not require a study of Oasis4 as a prerequisite.

We will revisit the paper with this in mind.

How does YAC ensure that the weights generated on the fly are of high quality? The text notes problems at poles in various algorithms, issues with different types of grid cells and different edge options, the requirement that one grid have either convex edges or be rectangular. Are those things checked on the fly? Are the weights somehow checked for conservative or gradient properties independently after they are generated? Are the properties at the pole checked? Is this a concern?

Detailed description of the interpolation algorithms themselves was not supposed to be part of the paper, because from our point of view they do not contain noteworthy differences to other implementations, except for the clipping that has its own paragraph. We will add a sentence to clarify our intention.

Page 1, Line 20: "not critical" is not a proper description. ESMs have been carrying out computations on moderate to high resolution grids for several years and the (pre) computation of interpolation weights in terms of both performance and quality has been a critical issue. This issue has taken up not insignificant resources in several projects including OASIS, ESMF, and SCRIP.

With this paragraph we refer to past generations of ESMs as stated in the beginning of this sentence. "Past generations" we used here as a synonym for coarse resolution models as it has been mainly used in the past CMIP phases. For those coarse resolution models with a low number of horizontal grid points of the order of less than 100.000, the compute time of the neighbourhood search has not been an issue, not in our models nor – to our knowledge – in other coupled models of these generations. We will clarify what we mean by "less critical" in the revised version of the manuscript by shifting the focus onto the horizontal resolution.

Section 3.2 describes the communication implementation and is far more important than much of the other material before it. I would like the description here to be expanded a bit, especially as related to lines 10-22. You need to describe "the non-blocking buffered send from Oasis4" for those that are not familiar. Please explain how the callback and data pointer work in a bit more detail. This is not entirely clear. Remember that your audience didn't work on either the Oasis4 or YAC implementation. Also, maybe there needs to be some further clarification on what aspect is being described. There is communication associated with weights generation, interpolation, and coupling data. Is this communication approach used for all of these?

We will remove the OASIS4 reference from this section. We will improve the description of the communication scheme and make clear that the communication scheme is used throughout the YAC internal workflow.

Section 3.4. The interpolation stack feature is well thought out and something that other weights generation methods are not able to do easily yet but is needed. Well done.

Thanks.

For us this part is more important than the interpolation methods themselves, because it might give others new ideas for future developments.

gmd-2015-267 comment to referee #2

Section 3.7. Does the current calculation of intersections guarantee conservation. The overlapping areas have to be computed in a way that the partial areas add up to the areas of each grid cell in total. I think page 14, line 3 suggests the areas are handled properly, but maybe a sentence stating this would clarify.

The sum of the partial areas will always add up to the area of the respective grid cell. Everything else is a bug or numerically inaccuracy. We might follow your suggestion and enhance the paragraph in this regard.

Section 4. I find the description of the user interface a bit out of place. The article is really focused on the interpolation weights generation. Nothing has been presented about how interpolation is carried out nor how models are coupled. This section, describing how to setup YAC via XML does not seem to fit into the paper.

Without a user interface the coupler library would be hard to use. Therefore we still think it is worthwhile to mention it, but we will move it to an Appendix.

Section 5. The total time and scaling for weights generation is very good. I think it's also important to show similar timing information for patch (1st, 2nd, and 3rd) and nearest neighbor calculations as well as bilinear if it's available. These are very different algorithms and I think providing scaling information for all three is important for this paper. In our experience, the 1st order conservative weights generation is sometimes the easiest and fastest to carry out. (This is also noted in Section 6, so data to back it up would be very useful)

Additional measurements will be added. The reading and writing of weight files works but is not yet optimised. Therefore we would like to refrain from adding detailed measurements. Furthermore, performance depends on various other external parameters like current workload of the system, the file system, the configuration of the IO library. We consider it far beyond the scope of this paper to analyse this in detail and provide sound interpretations of measured results.

*Figure 4 title seems incorrect. According to the text, it is the time to generate* 1*st* order conservative mapping weights, not to carry out remapping.

The title will be adjusted.

Section 6. Line 25. The only thing that has been shown in the text is the cost of the weights generation for 1st order conservative. It is a stretch to say "YAC scales reasonably well". Much of the YAC performance is not shown including remapping performance and cost to communicate data back and forth between models, the ability to support concurrency and other issues. gmd-2015-267 comment to referee #2

Except for the declaration of the local data on each process which contains no communication, the measurements show the whole initialisation cost.

We do not show measurements for the actual remapping and data exchange, because they typically have no impact on the performance of the model. Nevertheless, we will add respective measurements.

Figure 4 shows measurements with up to 12.288 MPI processes. In our opinion this shows the ability of YAC to support concurrency.

Section 6. Line 30. It would be nice to document the actual time needed to write and read mapping files as this is already available in YAC to have a quantitative comparison against the cost of generating the mapping weights. It would not surprise me if the read operation was more expensive than online weights generation for the test case used, and the actual numbers would add to the paper.

See our remark regarding additional measurements above.

Comments that are mainly on the coupler and not the paper:

Section 3.3. The requirement on the user to decompose the grid in a way that also includes a halo region and the rank owner of each halo gridpoint seems to be rather inflexible. That datatype and information will almost certainly have to be computed specifically to support the YAC weights generation interface and is unlikely to be a natural part of any model decomposition.

It is available in ICON. Furthermore, each model that uses advection and diffusion operators in a domain-decomposed world must have the knowledge about the halo. This includes ocean and atmosphere models. We agree that purely column based models like land components or chemistry (not chemistry transport) may have a problem here. In our case land and biogeochemistry are part of one component which does know about the composition. We do not see any need to modify the manuscript in this respect.

The halos are probably only needed in a subset of interpolation methods (like bilinear) and maybe YAC should be computing that connectivity, not the model. In addition, in a decomposition like round robin where each process has a random(ish) set of points, the halo description is going to require that "n" halo points be specified for each grid cell, increasing the memory and complexity. YAC would be much more usable if the connectivity were computed within the coupling layer when needed.

The halos are used to identify communication partners in the 1st-order conservative, patch recovery, nearest-neighbour and average interpolation. In addition it is used by

**gmd-2015-267 comment to referee #2**

the global search. The design is fundamentally based on having the halos. We could identify the owners of the halo points internally, but since ICON already provides us with that information, we did not yet see a need to do that. OASIS4 made an attempt to compute the connectivity within the coupling library. Our personal experience showed us that these attempts failed in the sense that is was not possible to provide a stable and performant algorithm. With every new grid configuration that was introduced to OASIS4 the algorithm needed to be revised. We do not see any need to modify the manuscript in this respect.

We do not mind not supporting round robin like decompositions.

Section 3.5. Do you have a bilinear interpolation option? This option is heavily used in ESMs.

Average with inverse distance weighting basically is linear interpolation. (page 12 line 8). The patch-recovery with a linear polynom fit or a 4-nearest-neighbour interpolation would be another alternative. For triangular grids a bilinear interpolation is not defined.

Section 8. The web info is pretty useful. Just FYI, I think it is missing a "getting started" type of documentation for new users to know what calls are needed and how to organize them in their models.

The source code contains trivial toy models that show how to use YAC. We will take up this suggestion and refer to them on the Doxygen page as well.

**Technical corrections**

We will consider all technical corrections and revise the text where requested by the reviewer.

Concerning references for both L'Huilier's Theorem and Girard's Theorem in the paper we consider this textbook material like the Pythagorean Theorem for which one usually does not provide citations. For both theorems we are not able to locate the original sources where these were published first by Simon Antoine Jean L'Huilier (1750 - 1840) and Albert Girard (1595 - 1632).

---

## Author Response (AR1)

gmd-2015-267 comment to referee #1

In red we repeat the reviewers remarks, in green we reply to the remarks, followed by sections and phrases in black (manuscript text in *italic black*) with changes that we applied to the discussion paper to obtain the revised version.

**Review #1**

We would like to thank the reviewer 1 for the careful reading and the detailed review.

**General Comments**

Overall, this manuscript is well written and very readable. It spends a lot of time discussing the capabilities of OASIS3 and OASIS4 outside of the introduction, and using them to justify decisions made for YAC. A lot of this seems unnecessary, and could easily be removed without negatively impacting the manuscript. Including this makes the manuscript seem almost like it is intended to describe those couplers as much as YAC.

In the "coupling" community, the question arose: why we had to invest into a new coupling software. Therefore, we felt the need to clarify the differences. In addition, OASIS3/4 is our main source of experience regarding coupling. Many of our decisions are based on these couplers, so use them to justify our design and give credit to them. As referee 2 had similar concerns we will shorten the manuscript and concentrate on YAC.

The performance results are overly simplistic. With only a single result provided for readers to gauge the performance of the coupler. While this may be the most expensive part of this coupler, it is also only performed once in a given simulation.

As stated in the paper, we did not really care about the performance of the coupler as long as it was within reasonable bounds. For the paper we put our focus on the design of the coupler itself. With the given performance results we wanted to show that it is working with acceptable performance. To enhance performance results we now provide additional measurements with other interpolation methods (see also our comment to referee 2).

It would be useful if the authors provided performance results for coupling steps including the interpolation from one grid to another and communicating the results.

A similar remark was made by referee 2. Our measurements showed that the coupling step itself (including interpolation and exchange) is really fast and hence boring. Furthermore, we do not consider the data exchange as time critical for real coupled applications. Wait time due to load imbalance is typically much more time consuming than the exchange of data. We now comment on the measurements for the remapping itself without adding a figure.

A new paragraph on the performance of the remapping step was added (see also response to referee 2):

*We also did measurements of the remapping step itself. For this we measured the maximum wall clock time required to do a bidirectional data exchange between both toy*

*models. The data exchange includes the exchange of halo data between source processes, the application of the remapping weights, and the transfer of the results to the target processes. All measurements varied in the range of 0.1ms to 10ms. The results have a mostly random pattern and no dependency on the number of nodes per component.*

Additionally, it would be useful to show a semi-realistic example where the two models that are coupled together are not collocated on the same nodes / processors.

It seems like the paper was not clear on this part. For example, in the last measurement we used 256 nodes (6.144 process/individual MPI ranks per component (see x-axis label of fig. 4) which adds up to a total of 512 nodes or 12.288 processes. Even though in each setup the two components are distributed over the same number of nodes each the processes are not collocated on the same nodes.

We added a sentence at the end of the second paragraph in section "Performance":

*Each component is run on its own set of nodes.*

It would also be useful to provide performance for the global search for additional interpolation methods, similar to the one-off discussion found in Section 6.

The global search is done only "*once for each pair of source and target grids for which any interpolation is required.*" (see section 3.3) The global search results are available to all interpolation methods, which in turn do additional communication to fulfil their special requirements.

One major issue with the manuscript is that it is not clearly documented where to gain access to the source code or find instructions for building / running the tests that are discussed in the manuscript. Navigating the Doxygen site (as recommended in the manuscript), I have been unable to find instructions for downloading / building YAC, so that part was not tested / reviewed, and I cannot comment on the readability of the source code.

As mentioned in section 8, on the main page of our Doxygen page (https://doc.redmine.dkrz.de/YAC/html/index.html) we have a paragraph about code availability:

**Code Availability**

*Tagged versions of the software are available upon request. Please contact Moritz Hanke ( hanke at dkrz.de ) or Rene Redler ( rene.redler at mpimpet.mpg.de ) and provide your name, institution and a few lines describing your intention with yac.*

In order to guarantee anonymity for the reviewers, we provided a link to an archive file to the Topical Editor. This tar file contains the status of our source code at the time when the paper was accepted for publication in GMDD. We are very sorry that this was not communicated clearly enough.

A README is shipped with the software which describes the basic steps to build the

library and run the tests.

**Corrections**

Page 10, figure 1: Please expand the caption of this figure. Alone the caption is not enough to determine what each of the subfigures represents, or what the triangles / rectangles are.

The figure caption has been expanded to:

*WOA09 January monthly mean sea surface salinity given on white rectangles interpolated to an ICON grid (triangles) using the YAC interpolation stack with 1st order conservative remapping (a) plus 1st-order polynomial fit (b) plus fixed values (c). (for further details see Sec. 2.4)*

Page 17, Figure 4 is captioned with…

The values in Fig. 4 (Fig 3 in the revised version) represent the wall clock time required for the call to yac_csearch." (see section 4, former section 5) This includes the global search and all communication and computation necessary to compute and distributed the weights required for the interpolation. The figure has been revised (see remarks above), text and captions were adjusted to the changes in the figure.

**Other corrections**

All corrections suggested by the referee have been applied to the revised version of the manuscript.

**Review #2**

Again we would like to thank the reviewer 2 for the careful reading and the constructive comments.

**General Overview**

I am a bit torn …

Here, R2 provided two alternatives to better set the focus of this paper. Unfortunately, we are not convinced by either of these. The first suggestion would lead us provide insight into the source code via a description of the internal subroutines and interfaces we we think is not doable and would probably go much beyond the scope of this paper. The second suggestion of giving a description about YAC would require major rewriting as R2 already stated. Furthermore it would distract the reader (and us, the authors) from what we would like to communicate. With this paper we wanted to present YAC with a special focus on a set of selected key aspects that we did differently compared to other coupling solutions. The weight generation algorithms we use are more or less the same as in any other coupler and are hence, from our point of view, sufficiently documented. To better set the focus of this paper we modified the title and abstract. Furthermore we added a sentence at the beginning of the last paragraph of the introduction.

New title: *YAC 1.2.0: New aspects for coupling software in Earth system modelling*

Abstract, added:   … *the new aspects* …

Introduction, last paragraph, added:

*With this publication we present YAC with a special focus on a set of selected key aspects that we do differently compared to other coupling solutions.*

**Specific Comments**

Overall, I believe there is a shortfall in the number of cited references.

We thank the referee for pointing us to missing references. Deliberately we do not provide a comprehensive overview on existing coupling software. Instead we already cite an existing review article (Valcke et al., 2012). In addition to that we now cite another review article by Valcke et al. (2012) on "Coupling technologies for Earth System Modelling". We have now added the missing references to coupling software that we mention explicitly, in particular MCT, OASIS3, and OASIS3-MCT. (References to ESMF are already provided.)

gmd-2015-267 comment to referee #2

For the interpolation methods we already provide the reference to the SCRIP library. Appropriate references for the patch recovery are are also given in the discussion paper. There is no point to provide references for a nearest-neighbour or average interpolation as this is standard textbook knowledge.

We added references:

Antonov, J., Seidov, D., Boyer, T., Locarnini, R. A., Mishonov, A., Garcia, H., Baranova, O., Zweng, M., and Johnson, D.: World Ocean Atlas 2009, Volume 2: Salinity. S. Levitus, Ed. NOAA Atlas NESDIS 69, U.S. Government Printing Office,Washington, D.C., 184 pages, 2010.

Jacob, R., Larson, J., and Ong, E.: MxN Communication and Parallel Interpolation in CCSM3 Using the Model Coupling Toolkit, Int. J. High Perf. Comp. App., 19, 293–307, 2005.

MPI Forum: MPI: A Message-Passing Interface Standard Version 3.1, Tech. rep., Knoxville, TN, USA, 2015.

Valcke, S., Balaji, V., Craig, A., DeLuca, C., Dunlap, R., Ford, R., Jacob, R., Larson, J., O'Kuinghttons, R., Riley, G., and Vertenstein, M.: Coupling technologies for Earth System Modelling, Geoscientific Model Development, 5, 1589–1596, doi:10.5194/gmd-5-1589-2012, http://www.geosci-model-dev.net/5/1589/2012/, 2012

Valcke, S.: The OASIS3 coupler: a European climate modelling community software, Geoscientific Model Development, 6, 373–388, doi:10.5194/gmd-6-373-2013, http://www.geosci-model-dev.net/6/373/2013/, 2013.

Valcke, S., Craig, T., and Coquart, L.: OASIS3-MCT User Guide, OASIS3-MCT 3.0, Tech. Rep. 1875, CERFACS/CNRS SUC URA, 2015.

Are the weights generated on the source or target side, on the union of all tasks or is it up to the user? Is remapping done on the source or target side or as part of communicating data between models? Is the communication of data between models separate from remapping or part of remapping?

The search and the calculation of weights is performed on all source processes (owners of the source grid). Here the data are interpolated onto the target before sending. In the discussion paper we already address this question on P11, L18. We have now introduced a separate section to explain this (see also next remark). We did not consider our implementation of the remapping step particularly interesting and therefore a detailed description has been omitted.

First, the source processes communicate with each other for a kind of halo exchange in order to provide the necessary data for the processes that have to calculate the

stencil. We follow the referees request and will modify the revised version of the manuscript accordingly by adding an additional paragraph.

The following paragraph was added:

*Once the interpolation is finished, each target point that can be interpolated has been assigned to a single source process. The source processes have the weights required to do the remapping of all target points assigned to them. The source processes might require data from other source processes to do the remapping of their target points. Therefore, in the actual remapping step, the respective data exchange between the source processes is performed first. Afterwards, the weights are applied to the source data in order to generate the target data, which is then sent to the target processes.*

The authors are extremely familiar with OASIS4, but it's unlikely the readers will be. It is fine to compare to OASIS4 and point readers to an OASIS4 reference, but it is also important to make sure the description does not assume the readers are inherently familiar with OASIS4 and that it does not require a study of OASIS4 as a prerequisite.

R1 made a similar remark. We have now removed the whole section about OASIS4 and kept only those references to OASIS4 where we refer to concepts that we inherited. Understanding YAC does not require studying OASIS4, and we are now convinced that this is not implied by our text anymore.

Deleted old section 2

The mentioning of OASIS4 has been removed from Sec. (YAC – Yet Another Coupler) or weakened.

P4 L 24ff, we removed:

P6, L22, we removed:

P7, L5, we replaced:
  **
by
  *based on our personal experience*

P15, L8ff: The whole paragraph has been removed.
We slightly rephrased other parts: P8, L11; P8, L20; P13, L17

How does YAC ensure that the weights generated on the fly are of high quality? The

text notes problems at poles in various algorithms, issues with different types of grid cells and different edge options, the requirement that one grid have either convex edges or be rectangular. Are those things checked on the fly? Are the weights somehow checked for conservative or gradient properties independently after they are generated? Are the properties at the pole checked? Is this a concern?

A detailed description of the interpolation algorithms themselves is not supposed to be part of the paper, because from our point of view they do not contain noteworthy differences to other implementations, except for the clipping that has its own paragraph.

We added a reference on first order conservative remapping:

*Jones, P. W.: First- and Second-Order Conservative Remapping Schemes for Grids in Spherical Coordinates, Monthly Weather Review, 127, 2204–2210, doi:10.1175/1520-0493(1999)127, http://dx.doi.org/10.1175/1520-0493(1999), 1999.*

Page 1, Line 20: "not critical" is not a proper description. ESMs have been carrying out computations on moderate to high resolution grids for several years and the (pre) computation of interpolation weights in terms of both performance and quality has been a critical issue. This issue has taken up not insignificant resources in several projects including OASIS, ESMF, and SCRIP.

With this paragraph we refer to past generations of ESMs as stated in the beginning of this sentence. "Past generations" we used here as a synonym for coarse resolution models as it has been mainly used in the past CMIP phases. For those coarse resolution models with a low number of horizontal grid points of the order of less than 100.000, the compute time of the neighbourhood search has not been an issue, not in our models nor – to our knowledge – in other coupled models of these generations. We have revised the whole paragraph to clarify this.

*We find that past generations of Earth system models with low spatial resolution and relatively simple block-structured grids, the neighbourhood search between any pair of grids, source and target, did not consume a significant amount of CPU time consumption. In the Coupled Model Intercomparison Project Phase 5 (CMIP5) the majority of coupled models were operated at horizontal resolution of 1 degree and more (see Appendix 9.A in Flato, 2013). Furthermore, the connectivity between neighbouring cells could be deduced by the coupler directly. In addition with the very low degree of parallelism, providing efficient algorithms to perform the neighbourhood search did not pose a significant challenge. With the advancement of new numerical models formulated on irregular grids, a trend towards very high resolution of more than 0.1 degree in the numerical grid and the high degree of parallelism with thousands of processes, we now require flexible and efficient algorithms also in the context of coupling.*

gmd-2015-267 comment to referee #2

Section 3.2 describes the communication implementation and is far more important than much of the other material before it. I would like the description here to be expanded a bit, especially as related to lines 10-22. You need to describe "the non-blocking buffered send from Oasis4" for those that are not familiar. Please explain how the callback and data pointer work in a bit more detail. This is not entirely clear. Remember that your audience didn't work on either the Oasis4 or YAC implementation. Also, maybe there needs to be some further clarification on what aspect is being described. There is communication associated with weights generation, interpolation, and coupling data. Is this communication approach used for all of these?

We have now explained the non-blocking buffered send

*This routine works similar to a MPI_Bsend but does not require to attach and detach message buffers prior to and after the call. Instead, the buffer management is handled internally.*

We improved the description of the communication scheme and now make clear that the communication scheme is used throughout the YAC internal workflow.

In the third paragraph of section 2.2 of the revised manuscript we replaced

*~~For the receiving counterpart we provide a modified version of the standard asynchronous receive operation. Instead of the request argument, our implementation requires a callback function and user-defined data pointer. This callback function is called once the respective message is received by the communication layer. The received data and the provided data pointer are passed as arguments to this function.~~*

with

*For the receiving counterpart we provide an asynchronous receive operation. In contrast to typical asynchronous MPI receive operations, our implementation has no request argument. Instead the user needs to provide a function pointer to a callback routine. A request for the receive is set up internally by the communication layer. Once the receive request is fulfilled, the communication layer will call the callback function associated to the request. The data that was received is passed as an argument to the callback function.*

We added a sentence at the end of section 2.3
*It is used for nearly all communication in the initialisation phase of YAC.*

Section 3.4. The interpolation stack feature is well thought out and something that other weights generation methods are not able to do easily yet but is needed. Well done.

Thanks.
For us this part is more important than the interpolation methods themselves, because it might give others new ideas for future developments.

Section 3.7. Does the current calculation of intersections guarantee conservation? The overlapping areas have to be computed in a way that the partial areas add up to the areas of each grid cell in total. I think page 14, line 3 suggests the areas are handled properly, but maybe a sentence stating this would clarify.

The sum of the partial areas will always add up to the area of the respective grid cell. Everything else is a bug or numerically inaccuracy. We might follow your suggestion and enhance the paragraph in this regard. We have added a sentence on page 14, line 4 to clarify this.

*The sum of the partial source cell areas will always add up to the area of the respective target grid cell up to numerical precision.*

Section 4. I find the description of the user interface a bit out of place. The article is really focused on the interpolation weights generation. Nothing has been presented about how interpolation is carried out nor how models are coupled. This section, describing how to setup YAC via XML does not seem to fit into the paper.

Without a user interface the coupler library would be hard to use. Therefore we still think it is worthwhile to mention it. We agree with the referee that the section is misplaced in the discussion paper.

Section 4 has been moved into the Appendix.

Section 5. The total time and scaling for weights generation is very good. I think it's also important to show similar timing information for patch (1st, 2nd, and 3rd) and nearest neighbor calculations as well as bilinear if it's available. These are very different algorithms and I think providing scaling information for all three is important for this paper. In our experience, the 1st order conservative weights generation is sometimes the easiest and fastest to carry out. (This is also noted in Section 6, so data to back it up would be very useful)

Additional measurements for patch 3rd and file base interpolation were added. The different interpolation methods show similar scaling behaviour. Therefore, we refrained from adding more measurements, as suggested by the reviewer, because it would not add more information about the scaling of YAC. We currently have some issues with the nearest neighbour interpolation and no time to investigate it that is why it is not included.

Added new diagram with additional measurements.

gmd-2015-267 comment to referee #2

Added description of measured interpolation methods:

*Three different interpolation methods were measured: first order conservative interpolation (conserv), patch recovery with a $3^{rd}$ order polynomial fit (patch), and a file based interpolation using the weights from the first order conservative interpolation (file).*

The description of what the measurements include was improved:

*The values in Fig. 3 represent the maximum wall clock time required for the call to yac_csearch, which is responsible for nearly all of the runtime consumed by YAC in the initialisation phase of the model run. This routine does the complete search, which is comprised of the interpolation-independent global search (see Sec. 2.3) and the interpolation-specific weight computation.*

The following sentence was removed, because it was no longer true for the latest measurements:

**

The description of the measurement graphs was simplified to avoid misunderstandings:

*For low number of nodes per component (up to 16) the computation required by the interpolation-independent global search and the interpolation-specific weight computation are the dominant factor for the runtime. This work scales well with the number of nodes, resulting in a minimum runtime of around half a second for the yac_csearch at 16 nodes or 384 processes per component. For higher node counts the computation time is negligible and the MPI communication contributes the preponderant part to the runtime.*

Interpretation of the differences between the three measured interpolation methods was added:

*The three measured interpolation methods generate different workloads, with patch recovery being the most time consuming one. With an increasing number of nodes per component the computation becomes less important for the total runtime of yac_csearch, the measurements of all three methods converge for high number of nodes as the communication dominates.*

Figure 4 title seems incorrect. According to the text, it is the time to generate $1^{st}$ order conservative mapping weights, not to carry out remapping.

The figure caption has been adjusted:

The new text reads as follows:

*Time required for global search and calculation of weights for different interpolation methods in both directions between an ICONR2B06 grid and a 405x405-cubed-sphere grid*

Section 6. Line 25. The only thing that has been shown in the text is the cost of the weights generation for 1st order conservative. It is a stretch to say "YAC scales reasonably well". Much of the YAC performance is not shown including remapping performance and cost to communicate data back and forth between models, the ability to support concurrency and other issues.

Except for the declaration of the local data on each process, which contains no communication, the measurements show the whole initialisation cost. This has now been clarified in the text and the title of the diagram (see changes listed above). In addition, measurements on the actual remapping step have been added. From our perspective this should cover the performance characteristics of YAC.
Figure 4 shows measurements with up to 12.288 MPI processes. In our opinion this shows the ability of YAC to support concurrency.

A new paragraph on the performance of the remapping step was added:
*We also did measurements of the remapping step itself. For this we measured the maximum wall clock time required to do a bidirectional data exchange between both toy models. The data exchange includes the exchange of halo data between source processes, the application of the remapping weights, and the transfer of the results to the target processes. All measurements varied in the range of 0.1ms to 10ms. The results have a mostly random pattern and no dependency on the number of nodes per component.*

Section 6. Line 30. It would be nice to document the actual time needed to write and read mapping files as this is already available in YAC to have a quantitative comparison against the cost of generating the mapping weights. It would not surprise me if the read operation was more expensive than online weights generation for the test case used, and the actual numbers would add to the paper.

Because weight files need to be written only once, we did not include any measurements on writing of them. The reading of weights files is part of the file based interpolation method and hence is now shown in the additional measurements in Figure 4.

Because the following paragraph was partially wrong and is covered by the additional measurements, it has been removed:

*As an alternative the user can precompute the interpolation weights in a preprocessing step and keep them in files such that the weights are read and*

*distributed by YAC at run-time. However, the parallel reading of the weight files and a* *proper distribution of the weights is not trivial and involves communication among the* *processes as well.*

**Comments that are mainly on the coupler and not the paper:**

We do not see any necessity to modify the text of the manuscript for these remarks.

Section 3.3. The requirement on the user to decompose the grid in a way that also includes a halo region and the rank owner of each halo gridpoint seems to be rather inflexible. That datatype and information will almost certainly have to be computed specifically to support the YAC weights generation interface and is unlikely to be a natural part of any model decomposition.

It is available in ICON. Furthermore, each model that uses advection and diffusion operators in a domain-decomposed world must have the knowledge about the halo. This includes ocean and atmosphere models. We agree that purely column based models like land components or chemistry (not chemistry transport) may have a problem here. In our case land and biogeochemistry are part of one component which does know about the composition. We do not see any need to modify the manuscript in this respect.

The halos are probably only needed in a subset of interpolation methods (like bilinear) and maybe YAC should be computing that connectivity, not the model. In addition, in a decomposition like round robin where each process has a random(ish) set of points, the halo description is going to require that "n" halo points be specified for each grid cell, increasing the memory and complexity. YAC would be much more usable if the connectivity were computed within the coupling layer when needed.

The halos are used to identify communication partners in the 1$^{st}$-order conservative, patch recovery, nearest-neighbour and average interpolation. In addition it is used by the global search. The design is fundamentally based on having the halos. We could identify the owners of the halo points internally, but since ICON already provides us with that information, we did not yet see a need to do that. OASIS4 made an attempt to compute the connectivity within the coupling library. Our personal experience showed us that these attempts failed in the sense that is was not possible to provide a stable and performant algorithm. With every new grid configuration that was introduced to OASIS4 the algorithm needed to be revised. We do not see any need to modify the manuscript in this respect.
We do not mind not supporting round robin like decompositions.

Section 3.5. Do you have a bilinear interpolation option? This option is heavily used in ESMs.

Average with inverse distance weighting basically is linear interpolation. (page 12 line

8). The patch-recovery with a linear polynomial fit or a 4-nearest-neighbour interpolation would be another alternative. For triangular grids a bilinear interpolation is not defined.

Section 8. The web info is pretty useful. Just FYI, I think it is missing a "getting started" type of documentation for new users to know what calls are needed and how to organize them in their models.

The source code contains trivial toy models that show how to use YAC. We will take up this suggestion and refer to them on the Doxygen page as well.

**Technical corrections**

The grammar is pretty good in this paper but could be improved …

For further improvement we prefer to rely on the professional service for the final version of the manuscript offered by the Copernicus team.

Please make sure to have references for both L'Huilier's Theorem and Girard's Theorem in the paper.

We consider this textbook material like the Pythagorean Theorem for which one usually does not provide citations. For both theorems we are not able to locate the original sources where these were published first by Simon Antoine Jean L'Huilier (1750 – 1840) and Albert Girard (1595 – 1632).

Please add a reference to the YAC Doxygen web site the first time it is mentioned in the paper. I know it is listed in section 8, but a reference would also be good.

The YAC Doxygen pages are first mentioned in Sec. 3.1 – Documentation on page 6, line 13 in the Discussion paper. There we already provide the URL as a footnote (bottom of page 6).

Other technical corrections

We have considered all other technical corrections and revised the text where requested by the reviewer.

[revised manuscript text omitted]

---

## Referee Report (RR1)

YAC 1.2.0: New aspects for coupling software in Earth system modelling.
M. Hanke, R. Redler, T. Holfeld, and M. Yastremsky.

This version of the paper is significantly improved.  The paper's focus is much improved
and many problems have been addressed.  However, the authors did not adequately
address the major shortcoming noted by both reviewers in the original paper,
lack of performance data.  In addition, there are a few new issues.

- Section 4 has several spelling and gramatical errors, that section should be
  reviewed carefully
  measurment -> measurement
  "The performance of the exchange step itself was in the tested setup good for all
node counts." needs to be written more clearly
  "The measurments suggest that the time for the exchange step is mainly influenced by
external factors like network utilisation of other applications running at the same
time." is something I think is incorrect.

- I think the new section discussing the remapping cost is probably not correct and
  needs to be updated.  That's page 15, lines 12-16.  The authors state performance
  was random and fell over 2 orders of magnitude.  I believe the performance should
  vary with nodes/cores and timing numbers should be reasonably reproducible.
  My sense, based on the statement in the paper, is that the timing may need to
  be redone, with appropriate barriers and/or a larger sample size.  You cannot
  measure one exchange in these kinds of tests, the time required is too small and
  is often shorter than the resolution of the timers.
  The point is that over the coarse of a long coupled run, coupling is done
  thousands or millions of times and it's that aggregate time that's important.
  At high resolution or high pe counts, the coupling time can be a bottleneck.
  To get an accurate representation of the time, the ping pong test should
  couple data back and forth for at least a few wall-seconds in total, maybe 1000s
  or 100s of thousands of times.  I would like to see a scaling curve, similar
  to Figure 3 for the ping pong test for a couple of resolutions that accurately
  reflect the communication/remapping cost.

  In addition, I think the time for remapping and the time for coupling could
  be separated in the presentation of the data.  It would be interesting to know how
  much of the ping pong time is spent remapping (including halo update and
  application of mapping weights) and how much is spent in the concurrent
  transfer of data.  There are cases when remapping is not needed.

- There is a set of timing data called "file" on figure 3
  but very little information is provided in the text how this is implemented (is it
  parallel I/O or "read and scatter" or something else) and there is no accompanying
  discussion.  The results are interesting and I feel I am left hanging.  I think
  some discussion needs to be added to discuss that set of data.  This is interesting
  data.

- Both reviewers felt the paper fell short on performance data and the paper
  has not substantially improved in that area.  As indicated above, some scaling
  curves for remapping and data transfer would make the paper significantly better.  I
  also think additional data should be provided on the weights generation, for
  additional resolutions and more mapping options.  The introduction
  also talked about how high resolution was the reason fast weights generation was
  important, but there are no high resolution results in the paper.  The authors

suggested moderate resolution is not interesting and no "significant challenge", but
then the authors only present results at moderate resolution.  I feel this
inconsistency needs to be addressed.

- Several of the authors' responses to the reviewers comments did not seem
  particularly well thought out.  Without focusing on that too much, I think there
  are a few important points that don't necessarily reflect on the current paper
  but might on YAC.

  First, a round robin or similar decomposition can be an excellent decomposition
  in cases where the cost per gridcell varies in time and/or space, like in radiation
  calculations (diurnal), sea ice coverage (seasonal), or land cover models (cost
  can vary diurnally and seasonally by location).  This decomposition probably
  deserves more consideration than "We do not mind not supporting round robin like
  decompositions"

  Second, the requirement to provide connectivity information as part of the
  weights generation is not state-of-the-art.  The author's response, "ICON can
  do this", "models with diffusion and advection operations will have the information
  readily available", and "it was too hard in OASIS4" are poor excuses.  If the tools
  in YAC are being designed to work generally, then other models will not be ICON,
  diffusion and advection operators may implicitly carry the connectivity information
  but accessing that information to pass to YAC might not be
  straight-forward.  In addition, there is no guarantee that the same halo is
  required for a diffusion or advection operation and for the YAC weights
  generation.  Finally, the fact that OASIS4 was not particularly successful
  in computing connectivity does not mean it's not possible.  Other weights generation
  tools are able to compute connectivity on the fly.

In summary, I believe the paper needs to be further revised to
 - review section 4 for grammar and spelling.
 - show accurate timing data for the ping pong test and if possible, to separate
   the remapping and coupling times into distinct terms.
 - add some discussion of the "file" data on figure 3.
 - extend the weights generation performance data further to include several
   additional mapping options including especially nearest neighbor.
 - provide timing results for the ping pong, file reading, and weights generation
   on at least one high resolution configuration as this is a major driver of the
   current implementation.

---

## Author Response (AR2)

Reply to review #3

YAC 1.2.0: New aspects for coupling software in Earth system modelling.
M. Hanke, R. Redler, T. Holfeld, and M. Yastremsky.

We would like to thank the referee for the concise and very constructive review. We understand the referees concerns. To a large extent we address the comments with the revised version of the manuscript. In black we copied the referees remarks and statements from the pdf file followed by our replies in blue, in *italic* we mark passages that we copied from the new revised version of the manuscript.

This version of the paper is significantly improved. The paper's focus is much improved and many problems have been addressed. However, the authors did not adequately address the major shortcoming noted by both reviewers in the original paper, lack of performance data. In addition, there are a few new issues.

Our intention for the paper was to present the aspects of YAC that make it different from other coupling solutions. Advertising YAC or its performance is still not our primary intention with this paper. To clarify this, we changed the sentence regarding the performance measurements in the abstract to:

*Preliminary performance measurements of a set of realistic use cases are presented to demonstrate the potential performance and scalability of our approach.*

Another similar sentence in the introduction was changes to:

*We present preliminary performance results for a set of realistic use cases in Sec. 3.*

In addition, we added the following paragraph to the performance section:

*In this section we present a first set of performance measurements to show the potential of YAC. As mentioned in Sec. 2.1, we do not yet concentrate on writing highly optimised code. Therefore, the measurements in this section are only preliminary. They can be seen as an estimate for the upper bound of the performance of YAC. In addition, scaling characteristics can be derived from them.*

We understand that the reviewer is interested in performance data of YAC. However, currently YAC is not at all optimized for performance. Therefore, the results only give general information on the scalability of our approach and the upper bound for the performance using idealised tests.

Nevertheless, in the latest version (2. revision) of the paper we added additional performance measurements and their analysis. The respective section is now the biggest one of the paper. In order to not further shift the focus of the paper, we are strongly against further extending this section. If the reviewer is still interested in even more performance data, we would like to refer the reviewer to the ESiWACE-project (https://www.esiwace.eu/).

- Section 4 has several spelling and gramatical errors, that section should be reviewed carefully

measurment -> measurement

"The performance of the exchange step itself was in the tested setup good for all node counts." needs to be written more clearly

"The measurments suggest that the time for the exchange step is mainly influenced by external factors like network utilisation of other applications running at the same time." is something I think is incorrect.

We have carefully checked section 4 (discussion and summary). Apart from the one spelling error already pointed out by the referee did not detect any further spelling errors. We use British English throughout the paper which is consistent with the Copernicus guidelines. Based on recommendations by a native speaker and as a consequence of the request by the reviewer to revise and add measurements we have revised this section, among them the two sentences pointed out by the referee. For brevity we have not included the revised section here as the details are accessible from the difference section below.

- I think the new section discussing the remapping cost is probably not correct and needs to be updated. That's page 15, lines 12-16. The authors state performance was random and fell over 2 orders of magnitude. I believe the performance should vary with nodes/cores and timing numbers should be reasonably reproducible.
  My sense, based on the statement in the paper, is that the timing may need to be redone, with appropriate barriers and/or a larger sample size. You cannot measure one exchange in these kinds of tests, the time required is too small and is often shorter than the resolution of the timers.
  The point is that over the coarse of a long coupled run, coupling is done thousands or millions of times and it's that aggregate time that's important. At high resolution or high pe counts, the coupling time can be a bottleneck.
  To get an accurate representation of the time, the ping pong test should couple data back and forth for at least a few wall-seconds in total, maybe 1000s or 100s of thousands of times. I would like to see a scaling curve, similar to Figure 3 for the ping pong test for a couple of resolutions that accurately reflect the communication/remapping cost.

  Indeed, in the first set of measurements we did too few exchanges per measurement. New and more accurate measurements have been added to the paper and accordingly the performance section has been completely revised. For brevity we have not included the revised section here as the details are accessible from the difference section below.

  In addition, I think the time for remapping and the time for coupling could be separated in the presentation of the data. It would be interesting to know how much of the ping pong time is spent remapping (including halo update and application of mapping weights) and how much is spent in the concurrent transfer of data. There are cases when remapping is not needed.

  Due to the asynchronous communication that we use for the data exchange between the source processes and the transfer of the remapping results from the source to the target processes, a clear separation and individual measurements of the steps would

not reflect the actual behaviour of YAC. As an alternative, we now provide additional numbers for our fixed interpolation, which is equivalent to a concurrent transfer of data between source and target – but without halo updates and without an application of mapping weights. (see page 15, line 20 ff)

- There is a set of timing data called "file" on figure 3 but very little information is provided in the text how this is implemented (is it parallel I/O or "read and scatter" or something else) and there is no accompanying discussion. The results are interesting and I feel I am left hanging. I think some discussion needs to be added to discuss that set of data. This is interesting data.

  The file interpolation mentioned in figure 3 is introduced earlier, but with a slightly different naming. To avoid confusion, we now use the same name for the interpolation throughout the text (file interpolation). Furthermore we have added some information about how the reading is done:

  *To avoid problems with memory consumption and runtime scalability, we use a parallel input scheme to read in the weight file. A subset of the source processes is selected to do the I/O. Each of these processes reads in an individual part of the data from the file. The data is then stored in a distributed directory (Pinar and Hendrickson, 2001). Afterwards, the source processes access the directory to get the data required to do the interpolation of the target points assigned to them.*

- Both reviewers felt the paper fell short on performance data and the paper has not substantially improved in that area. As indicated above, some scaling curves for remapping and data transfer would make the paper significantly better. I also think additional data should be provided on the weights generation, for additional resolutions and more mapping options. The introduction also talked about how high resolution was the reason fast weights generation was important, but there are no high resolution results in the paper. The authors suggested moderate resolution is not interesting and no "significant challenge", but then the authors only present results at moderate resolution. I feel this inconsistency needs to be addressed.

  We already provide the numbers for the weight generation for patch recovery, 1[st] order conservative remapping and the file interpolation. In the new revision measurements for the fixed interpolation were added.

  We would like to note again that the whole search and weight generation is implemented by making extensive use of asynchronous communication. As already mentioned for the ping pong exchange, artificially breaking the initialisation apart for more detailed timing results would not reflect the real behaviour of our approach. We prefer to only show performance numbers that are relevant for the end-users of YAC. However, the newly added measurements for the fixed interpolation allow a slightly deeper analysis when compared with the other interpolation methods (see revised sections about performance and discussion).

  As requested by the reviewer we added measurements for an additional grid configuration. With ~22 km in the atmosphere and ~12 km in the ocean it uses grids

with resolutions two times higher than the original ones.

In the summary (see below) the referee explicitly asks for performance numbers for the nearest neighbour interpolation. YAC supports nearest neighbour interpolation. The nearest neighbour interpolation is listed in the paper as it is part of the functionality of YAC and is working correctly. However, we still see significant performance issues with this interpolation, which are already addressed in the original version of the manuscript. Thus the work on this interpolation is not yet finalised. We strongly prefer not to show performance numbers in a publication for an algorithm, which is in some sense still under construction.

- Several of the authors' responses to the reviewers comments did not seem particularly well thought out. Without focusing on that too much, I think there are a few important points that don't necessarily reflect on the current paper but might on YAC.

  First, a round robin or similar decomposition can be an excellent decomposition in cases where the cost per gridcell varies in time and/or space, like in radiation calculations (diurnal), sea ice coverage (seasonal), or land cover models (cost can vary diurnally and seasonally by location). This decomposition probably deserves more consideration than "We do not mind not supporting round robin like decompositions"

  We apologise for having been too superficial in our first response. We fully agree with the referee that a round robin or similar decomposition does have advantages. Nevertheless, we do not claim that YAC is already complete. We prioritise our work such that we are able to support those applications for which we would like to use our software first and possibly add more functionality when required at a later stage.

  Actually, with some limitations these types of decomposition are supported. The file interpolation makes no use of the halo cells. With this interpolation method YAC could theoretically support this type of decomposition. However, we did not yet test such a use case.

  Second, the requirement to provide connectivity information as part of the weights generation is not state-of-the-art. The author's response, "ICON can do this", "models with diffusion and advection operations will have the information readily available", and "it was too hard in OASIS4" are poor excuses. If the tools in YAC are being designed to work generally, then other models will not be ICON, diffusion and advection operators may implicitly carry the connectivity information but accessing that information to pass to YAC might not be straight-forward. In addition, there is no guarantee that the same halo is required for a diffusion or advection operation and for the YAC weights generation. Finally, the fact that OASIS4 was not particularly successful in computing connectivity does not mean it's not possible. Other weights generation tools are able to compute connectivity on the fly.

  We fully agree with the referee that the computation of connectivity on the fly is possible. What we tried to say in our first reply was – and obviously we failed in this first attempt – that we find it not straightforward to perform such a computation in a fully parallel, efficient, and robust way. At least this was the lesson that we learned

from OASIS4, and with the first version of YAC we preferred to bypass this complexity and invest our time and effort on other – in our very personal opinion more interesting – functionality like the interpolation stack.

In our first reply we already stated that we see an issue with pure column-based model components like e.g. land models. For the neighbourhood search we currently do require a halo with a width of at least one cell around the compute domain. We use this halo solely to identify neighbouring processes. The stencils used for the interpolation are not limited by the halo and can even go across multiple compute domains. Our point is that most applications that operate on dynamical kernels are able to provide the halo information that YAC requires, and the applications that we have worked with so far are able to provide this halo either directly or with very minor effort.

As the referee states, these remarks do not apply to the manuscript but more to aspects of the software itself. Therefore, we have not changed the manuscript to further discuss theses aspects.

In summary, I believe the paper needs to be further revised to

- review section 4 for grammar and spelling.

  We have revised section 4.

- show accurate timing data for the ping pong test and if possible, to separate the remapping and coupling times into distinct terms.

  We have revised the timing data for the ping pong test. We did not explicitly separate remapping and coupling, but added measurements for the fixed interpolation, which basically is an exchange without remapping.

- add some discussion of the "file" data on figure 3.

  We cleaned up the wording to ensure that we use identical phrases throughout the text for this type of interpolation. The technical implementation of the reading of a weight file and the distribution of the weights to the source processes is now described. We compare the file interpolation with the online computation of weights for the conservative remapping.

- extend the weights generation performance data further to include several additional mapping options including especially nearest neighbor.

  Above we provide our reasons why we prefer not to show performance data for the nearest neighbour interpolation. We already provide performance data for the most important and quite costly interpolations (patch recovery, 1$^{st}$ order conservative remapping, file). In addition, data for the fixed interpolation was added. We are convinced that these numbers provide a firm estimate on the scalability of the approach used in YAC and on the upper bound for the absolute performance that is to be expected from YAC.

Recent changes to our testing system led to changes in the performance data. Especially for high node counts the results are now better compared to the measurements provided in the previous revision of the paper.

- provide timing results for the ping pong, file reading, and weights generation on at least one high resolution configuration as this is a major driver of the current implementation.

We added timing results for a two times higher horizontal resolution (~22 km in the atmosphere and ~12 km in the ocean) and discuss them. As the benchmark runs occupy a significant portion of the whole system at DKRZ, our ability to perform more benchmark tests in near future is limited. We would like to note again that performance is not the main topic of our paper. Nevertheless, we are convinced that the additional performance results now provide a good estimate for the scaling behaviour of YAC and what we can expect from YAC when applied to even higher resolutions.

In summary, as a consequence the performance and discussion sections have been substantially revised.

[revised manuscript text omitted]